# MixLinear: Extreme Low Resource Multivariate Time Series Forecasting with $0.1K$ Parameters

## Abstract

Recently, there has been a growing interest in Long-term Time Series Forecasting (LTSF), which involves predicting long-term future values by analyzing a large amount of historical time-series data to identify patterns and trends. There exist significant challenges in LTSF due to its complex temporal dependencies and high computational demands. Although Transformer-based models offer high forecasting accuracy, they are often too compute-intensive to be deployed on devices with hardware constraints. On the other hand, linear models aim to reduce the computational overhead by employing decomposition methods in the time domain or compact representations in the frequency domain. In this paper, we propose MixLinear, an ultra-lightweight multivariate time series forecasting model specifically designed for resource-constrained devices. MixLinear effectively captures both temporal and frequency domain features by modeling intra-segment and inter-segment variations in the time domain and extracting frequency variations from a low-dimensional latent space in the frequency domain. By reducing the parameter scale of a downsampled $n$-length input/output one-layer linear model from $O(n^2)$ to $O(n)$, MixLinear achieves efficient computation without sacrificing accuracy. Extensive evaluations with four benchmark datasets show that MixLinear attains forecasting performance comparable to, or surpassing, state-of-the-art models with significantly fewer parameters ($0.1K$), which makes it well-suited for deployment on devices with limited computational capacity.

## 1 Introduction

Time-series modeling is crucial for various fields, including climate science (Moon & Wettlaufer, 2017), biological research (Watson et al., 2021), medicine (Kim et al., 2014), retail (Nunnari & Nunnari, 2017), and finance (Sezer et al., 2020). Accurate time series forecasting is essential for informed decision-making and strategic planning in these domains. Traditional approaches, such as Autoregressive (AR) models (Nassar et al., 2004), exponential smoothing (Gardner Jr, 1985), and structural time-series models (Harvey, 1990), have established a strong foundation for time-series forecasting. In recent years, there has been a growing interest in Long-term Time Series Forecasting (LTSF), which aims to predict long-term future values by identifying patterns and trends in large amounts of historical time-series data. Recent research has demonstrated that leveraging advanced machine learning techniques, such as Gradient Boosted Regression Trees (GBRT) (Mohan et al., 2011), and deep learning models, including Recurrent Neural Networks (RNN) (Salehinejad et al., 2017) and Temporal Convolutional Networks (TCN) (He & Zhao, 2019), yields significant performance improvements over traditional methods.

Over the last few years, significant efforts have been made to explore the use of Transformers for LTSF and produced many good models, such as LogTrans (Nie et al., 2022), Informer (Zhou et al., 2021), Autoformer (Wu et al., 2021), Pyraformer (Liu et al., 2021), Triformer (Cirstea et al., 2022), FEDformer (Zhou et al., 2022b), and PatchTST (Nie et al., 2023). Those models achieve good forecasting performance at the cost of introducing significant computation overhead due to the use of the self-attention mechanism, which scales quadratically with sequence length $L$. The high computational demands and large memory requirements of these models hinder their deployment for LTSF tasks on resource-constrained devices. To address this limitation and facilitate low-resource usage,

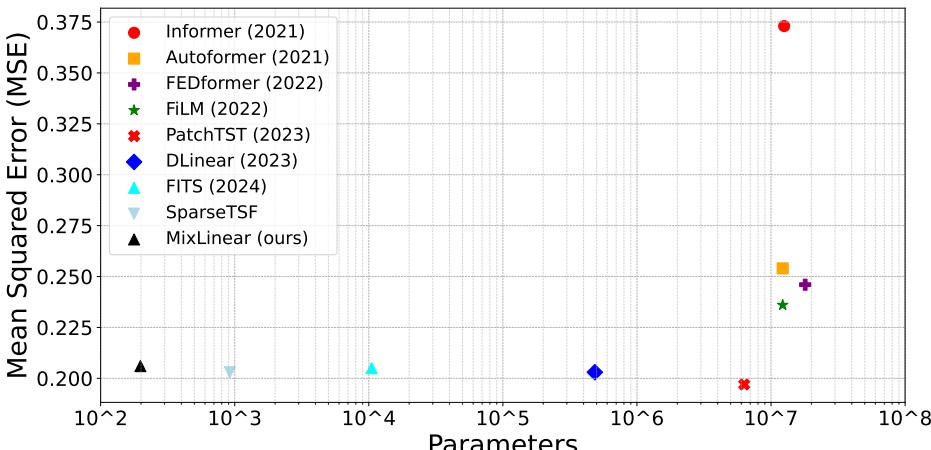

Figure 1: Comparison of MSE and parameters between MixLinear and other mainstream models of the Electricity dataset with a forecast horizon of 720.

researchers have proposed refined linear models based on decomposition techniques that achieve comparable performance with significantly fewer parameters. For instance, FITS (Xu et al., 2024) attains superior performance using interpolation in the complex frequency domain with only $10K$ parameters, while SparseTSF (Lin et al., 2024) further reduces the parameter count to $1K$ while maintaining robust performance.

However, current research in LTSF focuses on efficiently decomposing and capturing dependencies from either the time domain or frequency domain. For instance, Informer employs an attention-distilling method to reduce complexity (Zhou et al., 2021) in the time domain, PatchTST utilizes a patching technique to transform time series into subseries-level patches for increased efficiency (Nie et al., 2023), and SparseTSF simplifies forecasting by decoupling periodicity and trend (Lin et al., 2024). In the frequency domain, FEDformer decomposes sequences into multiple frequency domain modes using frequency transforms to extract features (Zhou et al., 2022b). TimesNet (Wu et al., 2022) employs a frequency-based method to separate intraperiod and interperiod variations. FITS utilizes a complex-valued neural network to capture both amplitude and phase information simultaneously, providing a more comprehensive and efficient approach to processing time series data (Xu et al., 2024).

In this paper, we introduce MixLinear a highly lightweight multivariate time series forecasting model, which efficiently captures the temporal and frequency features from both time and frequency domains. It captures intra-segment and inter-segment variations in the time domain by decoupling channel and periodic information from the trend components, breaking the trend information into smaller segments. In the frequency domain, it captures frequency domain variations by mapping the decoupled time series subsequences (trend) into a latent frequency space and reconstructing the trend spectrum. MixLinear reduces the parameter requirement from $O(n^2)$ to $O(n)$ for $L$-length inputs/outputs with a known period $w$ and subsequence length $n = \left\lceil \frac{L}{w} \right\rceil$. Our comprehensive evaluation of LTSF with benchmark datasets shows that MixLinear provides comparable or better forecasting accuracy with much fewer parameters ($0.1K$) compared to state-of-the-art models. For instance, as Fig. 1 shows, MixLinear achieves a Mean Squared Error (MSE) of $0.208$ on the Electricity dataset [1] with a forecast horizon of 720 with 195 parameters.

In summary, our contributions in the paper are as follows:

- We introduce an extremely lightweight model MixLinear that can achieve state-of-the-art comparable or better forecasting accuracy with only $0.1K$ parameters.

---

[1]The Electricity dataset contains the hourly electricity consumption of 321 customers, spanning the period from 2012 to 2014.

- To our knowledge, Mixlinear is the first lightweight LTSF model that captures temporal and frequency features from both time and frequency domains. MixLinear applies the trend segmentation in the time domain to capture the intra-segment and inter-segment variations. MixLinear captures amplitude and phase information by reconstructing the trend spectrum from low dimensional latent space in the frequency domain.

- To evaluate our model, we conduct experiments on several widely used LTSF benchmark datasets. MixLinear consistently delivers top-tier performance across a variety of time series tasks and achieves up to a $5.3\%$ reduction in MSE on these benchmarks.

## 2 PRELIMINARIES

**Long-term Time Series Forecasting.** The task of LTSF involves predicting future values over an extended horizon using previously observed multivariate time series data. It is formalized as $\hat{x}_{t+1:t+H} = f(x_{t-L+1:t})$, where $x_{t-L+1:t} \in \mathbb{R}^{L \times C}$ and $\hat{x}_{t+1:t+H} \in \mathbb{R}^{H \times C}$. In this formulation, $L$ denotes the length of the historical observation window, $C$ represents the number of distinct features or channels, and $H$ denotes the length of the forecast horizon. The main goal of LTSF is to extend the forecast horizon $H$ as it provides rich and advanced guidance in practical applications. However, an extended forecast horizon $H$ often requires more parameters and significantly increases the parameter scale of the forecasting model.

**Lightweight Time Series Forecasting.** Recently, there has been a growing interest in developing lightweight models for LTSF. DLinear (Zeng et al., 2023) demonstrates that simple linear models can effectively capture temporal dependency and outperform transformer-based models. DLinear shares the weights across different variates, does not model spatial correlations, and transforms the multivariate input $x_{t-L+1:t} \in \mathbb{R}^{L \times C}$ to the output $\hat{x}_{t+1:t+H} \in \mathbb{R}^{H \times C}$ by reformulating it into a univariate mapping $x_{t-L+1:t} \in \mathbb{R}^{L}$ to $\hat{x}_{t+1:t+H} \in \mathbb{R}^{H}$. On the other hand, FITS (Xu et al., 2024) employs a harmonic content-based cutoff frequency selection method that reformulates the univariate input $x_{t-L+1:t} \in \mathbb{R}^{L}$ to output $\hat{x}_{t+1:t+H} \in \mathbb{R}^{H}$ by mapping it to the frequency domain and reduces the input length from $L$ to $n^{COF}$, where $n^{COF}$ is the cutoff frequency and $n^{COF} << L$. FITS significantly reduces the parameter scale (from $140K$ to $10K$). SparseTSF takes a different approach by decoupling periodicity and trend components in time series data through aggregation and downsampling and reformulates the univariate input $x_{t-L+1:t} \in \mathbb{R}^{L}$ to output $\hat{x}_{t+1:t+H} \in \mathbb{R}^{H}$ by mapping the trend component $x_{t-n+1:t}$ to $\hat{x}_{t+1:t+m}$, where $n = \lceil \frac{L}{w} \rceil$, $m = \lceil \frac{H}{w} \rceil$, and $w$ is the period. SparseTSF reduces the parameter scale to as low as $1K$.

## 3 MIXLINEAR

### 3.1 OVERVIEW

Current research in LTSF focuses on efficiently decomposing and capturing temporal dependencies from the time or frequency domain. The key innovation of MixLinear lies in its ability to extract features from both domains while minimizing the number of neural network parameters. However, combining time and frequency domain models can significantly increase the parameter scale. Mix-Linear addresses such an issue by substantially reducing the parameter count without compromising prediction performance. Figure 2 illustrates the overall architecture of MixLinear, which consists of two key processes: Time Domain Transformation and Frequency Domain Transformation. Unlike the existing linear models that apply pointwise transformations, our Time Domain Transformation captures inter-segment and intra-segment dependencies by splitting the decoupled time series (trend) into segments. Such a method significantly reduces the model parameter scale and enhances the locality, which is unavailable in the pointwise methods. In contrast to the existing frequency-based models that perform transformation on the entire series, our Frequency Domain Transformation focuses on transforming more compact trend components in a lower-dimensional latent space, which reduces the model complexity by learning frequency variations more effectively. The overview workflow of MixLinear can be found in Appendix A.1.

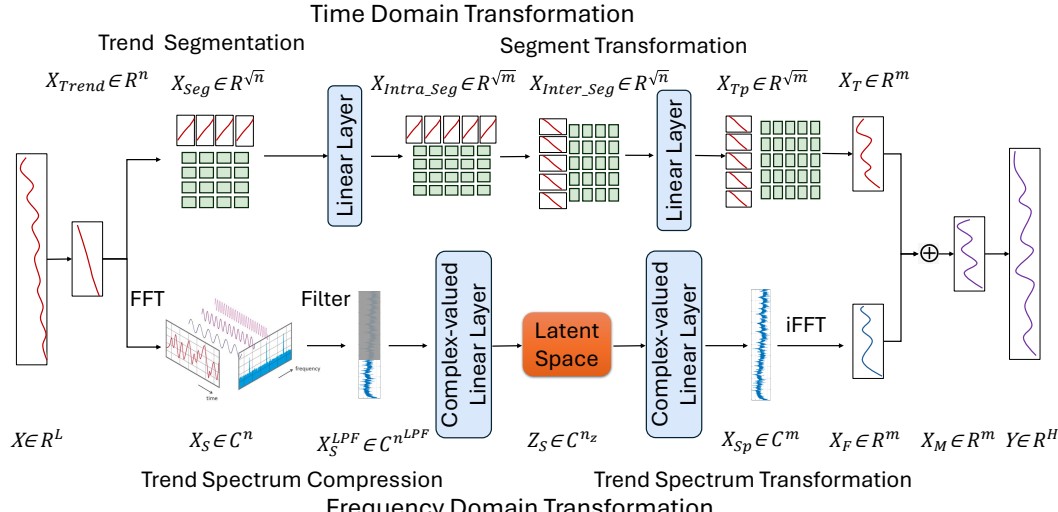

Figure 2: Architecture of MixLinear. MixLinear first extracts the trend information by downsampling the time series with a period of $w$. In the time domain, it divides the trend into segments and applies two linear transformations: one to capture intra-segment dependencies and the other to capture inter-segment dependencies. In the frequency domain, it performs the Fast Fourier Transform (FFT) to project the data into the frequency domain, followed by a low pass filter and two complex-valued linear layers for spectral compression and reconstruction. The inverse FFT (iFFT) is then used to revert the data back to the time domain. Finally, the outputs from both time and frequency domains are merged and the data is upsampled by the same period $w$.

## 3.2 TIME DOMAIN TRANSFORMATION

The existing lightweight linear models, such as SparseTSF (Lin et al., 2024), decouple the periodic and trend components and apply a pointwise linear transformation to the trend components. In contrast, Time Domain Transformation in MixLinear divides the trend components into smaller segments and applies two linear transformations to capture intra-segment and inter-segment dependencies. Such an approach significantly reduces the model complexity while enhancing the locality of the model which is not available at the point level (Nie et al., 2023). Time Domain Transformation includes two main subprocesses: Trend Segmentation and Segment Transformation.

**Trend Segmentation.** Given the time series data $X \in \mathbb{R}^L$ with the period $w$, we perform aggregation and downsampling to extract the trend components (Lin et al., 2024). For aggregation, we apply a 1D convolution with a kernel size of $w$, which allows us to aggregate all the information within each period at every time step. We then downsample the aggregated series by the period $w$, resulting in the trend component $X_{\text{Trend}} \in \mathbb{R}^n$, where $n = \lceil \frac{L}{w} \rceil$. This method effectively decouples the periodic and trend components, providing a more compact representation in which each trend time point encapsulates all the information from one period in the original series. Also, we do zero padding to $X_{\text{Trend}}$ to make $\sqrt{n}$ to be an integer. And then we split the trend components $X_{\text{Trend}} \in \mathbb{R}^n$ into smaller trend segments $X_{\text{Seg}} \in \mathbb{R}^{\sqrt{n}}$. We select the segment length as $\sqrt{n}$ to minimize the parameter size.

**Segment Transformation.** Segment Transformation begins by applying a linear layer to trend segments $X_{\text{Seg}}$ to capture intra-segment dependencies. This produces the intra-segment prediction $X_{\text{Intra\_Seg}} \in \mathbb{R}^{\sqrt{m}}$, where $m = \lceil \frac{H}{w} \rceil$ and $H$ is the forecast horizon. $X_{\text{Intra\_Seg}}$ is then upsampled by $\sqrt{n}$, transposed, and downsampled by $\sqrt{m}$ to obtain the inter-segment series $X_{\text{Inter\_Seg}} \in \mathbb{R}^{\sqrt{n}}$. Another linear layer is applied to the inter-segment series $X_{\text{Inter\_Seg}}$ to obtain the inter-segment prediction $X_{\text{Tp}} \in \mathbb{R}^{\sqrt{m}}$. Finally, $X_{\text{Tp}}$ is upsampled by $\sqrt{m}$ to produce the time-domain output $X_{\text{T}} \in \mathbb{R}^m$. Leveraging Segment Transformation, MixLinear reduces the model complexity from

$n \times m$ to $2 \times \sqrt{n} \times \sqrt{m}$. When $m = n$, this method offers a significant reduction in complexity (from $O(n^2)$ to $O(n)$). In addition, both inter-segment and intra-segment variations are captured.

## 3.3 FREQUENCY DOMAIN TRANSFORMATION

The frequency domain representation of time series data promises a more compact and efficient portrayal of inherent patterns (Xu et al., 2024). Unlike FITS, MixLinear applies frequency domain transformation to downsampled time series subsequences (trend) and learns the frequency feature from latent space which focuses on the important bits of the data and trains in a lower dimensional, computationally much more efficient space (Rombach et al., 2022). It has two subprocesses: Trend Spectrum Compression and Trend Spectrum Transformation.

**Trend Spectrum Compression.** Given the time series data $X \in \mathbb{R}^L$ with the period $w$, we first decompose the trend components to get $X_{Trend} \in \mathbb{R}^n$, where $n = \lceil \frac{L}{w} \rceil$. Then we apply FFT to the trend components $X_{Trend}$ and convert it into the frequency domain. The FFT computation for a discrete sequence $\{x_k\}_{k=0}^{n-1}$ is given by:

$$\boldsymbol{X}_{\mathrm{S}}[k] = \sum_{m=0}^{n-1} x_m \cdot e^{-j \frac{2\pi}{n} km}, \tag{1}$$

where $j$ is the imaginary unit, $k$ is the frequency index, and $m$ is the time index (Xu et al., 2024). $X_S \in \mathbb{C}^n$ is a complex-valued representation that concisely encapsulates the amplitude and phase of each frequency component in the Fourier domain. This transformation effectively converts the time-domain sequence into its frequency-domain representation, which captures key amplitude and phase features.

Next, we apply a Low-Pass Filter (LPF) to $X_S$ to remove the high-frequency components typically associated with noise and preserve the lower-frequency components that are more relevant for forecasting (Xu et al., 2024). A specified cutoff threshold is used to discard high-frequency components. LPF converts the complex-valued spectral $X_S \in \mathbb{C}^n$ to $X_S^{LPF} \in \mathbb{C}^{n^{LPF}}$, where $n^{LPF}$ is the cutoff frequency threshold, which is smaller than $n$.

Finally, MixLinear compresses the filtered spectral representation, $X_S^{LPF}$, into a lower-dimensional latent space. Specifically, a complex-valued linear layer is applied to the filtered spectral data to obtain the latent frequency space representation, $Z_S \in \mathbb{C}^{n_z}$.

**Trend Spectrum Transformation.** Trend Spectrum Transformation reconstructs the trend spectrum from the latent space, and transforms it back to its original form by upsampling. The process applies a complex-valued linear layer to the latent space representation $Z_S \in \mathbb{C}^{n_z}$, transforms it into the spectrum $X_{Sp} \in \mathbb{C}^m$. Such a transformation is achieved through the operation $X_{Sp} = W \cdot Z_S + b$, where $W$ is a complex-valued weight matrix and $b$ is a bias term.

Once the spectrum $X_{Sp}$ is obtained, the iFFT is applied to convert the spectrum back to the time domain. The iFFT is mathematically defined as:

$$X_F(n) = \frac{1}{m} \sum_{k=0}^{m-1} X_{Sp}(k) e^{i2\pi kn/m}, \tag{2}$$

where $m$ is the length of the spectrum, $X_{Sp}(k)$ represents the frequency domain values for each frequency $k$, and $e^{i2\pi kn/m}$ is the complex exponential term used to translate frequency components back into the time domain. This operation results in the time-domain signal $X_F \in \mathbb{R}^m$, which represents the trend prediction in the frequency domain.

The total parameter size used in the frequency domain is $(m + n) * n_z$. When $m = n$. the total parameter required becomes $2n * n_z$. As $n_z \ll n$, those two linear transformations reduces the parameter scale from $O(n^2)$ to $O(n)$. We set $n_z$ to 2 in the experiment section to reduce the parameter scale as much as possible.

# 4 EXPERIMENT

In this section, we first outline our experimental setup. We then compare MixLinear with the baseline models and assess its effectiveness in achieving high forecasting accuracy with minimal parameters by integrating both time and frequency domain features. Lastly, we evaluate the generalization capability of MixLinear. Detailed analyses of MixLinear's performance in ultra-long period scenarios, as well as the impact of the low-pass filter cutoff frequency, are provided in the appendix.

## 4.1 EXPERIMENT SETUP

**Datasets.** We perform experiments with four benchmark LTSF datasets (i.e., ETTh1, ETTh2, Electricity, and Traffic) that exhibit daily periodicity. The ETTh1 and ETTh2 datasets contain hourly data collected from Informer (Zhou et al., 2021). The Electricity dataset contains the hourly electricity consumption of 321 customers from the University of California, Irvine Machine Learning Repository website. The Traffic dataset is a collection of hourly data from the California Department of Transportation. The Solar dataset records the solar power production from 137 PV plants in Alabama State in 2016. The Exchange dataset collects the daily exchange rates of 8 foreign countries. More details about those datasets can be found in Appendix A.2.

**Baselines.** We conduct a comparative analysis of MixLinear against state-of-the-art baselines in the field, including FEDformer (Zhou et al., 2022b), TimesNet (Wu et al., 2022), SCINet (Liu et al., 2022), ITransformer (Liu et al., 2023), and PatchTST (Nie et al., 2023). In addition, we compare MixLinear against three lightweight models: DLinear (Zeng et al., 2023), FITS (Xu et al., 2024), and SparseTSF (Lin et al., 2024). More details about those baselines can be found in Appendix A.3.

**Environment.** MixLinear and our baselines are implemented using PyTorch (Paszke et al., 2019). All experiments are performed on a single NVIDIA A100 GPU with $80GB$ of memory. More details on our experimental setup are presented in Appendix A.4.

## 4.2 PREDICTION PERFORMANCE

Table 1: MSE results of multivariate long-term time series forecasting comparing MixLinear against baselines. The top three results are highlighted in **bold**. The best results are in **bold** and underlined. "Diff." represents the difference in MSE prediction performance between MixLinear and the second-best/best result baseline, with positive values indicating performance improvement.

| Models | | MixLinear | SparseTSF | FITS | DLinear | PatchTST | iTransformer | SCINet | TimesNet | FEDformer | Diff. |
|---|---|---|---|---|---|---|---|---|---|---|---|
| Data | Horizon | (ours) | (2024) | (2024) | (2023) | (2023) | (2023) | (2022) | (2022) | (2022) | |
| ETTh1 | 96 | **0.351** | **0.362** | 0.382 | 0.384 | 0.385 | 0.386 | **0.375** | 0.384 | **0.375** | +0.011 |
| | 192 | **0.395** | **0.403** | 0.417 | 0.443 | **0.413** | 0.441 | 0.429 | 0.436 | 0.427 | +0.008 |
| | 336 | **0.411** | **0.434** | **0.436** | 0.446 | 0.440 | 0.487 | 0.504 | 0.491 | 0.459 | +0.023 |
| | 720 | **0.423** | **0.426** | **0.433** | 0.504 | 0.456 | 0.503 | 0.544 | 0.521 | 0.484 | +0.003 |
| ETTh2 | 96 | 0.283 | 0.294 | **0.272** | **0.282** | **0.274** | 0.297 | 0.289 | 0.340 | 0.340 | -0.011 |
| | 192 | **0.336** | 0.339 | **0.333** | 0.340 | **0.338** | 0.380 | 0.372 | 0.402 | 0.433 | -0.003 |
| | 336 | **0.355** | 0.359 | **0.355** | 0.414 | 0.367 | 0.428 | **0.365** | 0.452 | 0.508 | +0.004 |
| | 720 | **0.380** | 0.383 | **0.378** | 0.588 | 0.391 | 0.427 | 0.475 | 0.462 | 0.480 | -0.002 |
| Electricity | 96 | **0.138** | **0.138** | 0.145 | **0.140** | 0.129 | 0.148 | 0.168 | 0.168 | 0.188 | -0.009 |
| | 192 | 0.154 | **0.151** | 0.159 | **0.153** | **0.149** | 0.162 | 0.175 | 0.184 | 0.197 | -0.005 |
| | 336 | **0.170** | **0.166** | 0.175 | **0.169** | **0.166** | 0.178 | 0.189 | 0.198 | 0.212 | -0.004 |
| | 720 | **0.209** | **0.205** | 0.212 | **0.204** | 0.210 | 0.225 | 0.231 | 0.220 | 0.244 | -0.005 |
| Traffic | 96 | **0.389** | **0.389** | 0.398 | 0.413 | **0.366** | 0.395 | 0.613 | 0.593 | 0.573 | -0.023 |
| | 192 | **0.403** | **0.398** | 0.409 | 0.423 | **0.388** | 0.417 | 0.535 | 0.617 | 0.611 | -0.015 |
| | 336 | **0.416** | **0.411** | 0.421 | 0.437 | **0.398** | 0.433 | 0.540 | 0.629 | 0.621 | -0.018 |
| | 720 | **0.452** | **0.448** | 0.457 | 0.466 | **0.457** | 0.467 | 0.620 | 0.640 | 0.630 | -0.004 |
| Solar | 96 | **0.211** | **0.211** | **0.195** | 0.290 | 0.265 | **0.203** | 0.237 | 0.373 | 0.286 | -0.017 |
| | 192 | **0.227** | **0.225** | **0.216** | 0.320 | 0.288 | **0.233** | 0.280 | 0.397 | 0.291 | -0.011 |
| | 336 | **0.240** | **0.241** | **0.232** | 0.353 | 0.301 | **0.248** | 0.304 | 0.420 | 0.354 | -0.008 |
| | 720 | **0.240** | **0.241** | **0.242** | 0.357 | 0.295 | 0.249 | 0.308 | 0.420 | 0.380 | +0.001 |
| Exchange | 96 | **0.088** | 0.105 | **0.086** | **0.087** | 0.087 | **0.086** | 0.267 | 0.107 | 0.148 | -0.002 |
| | 192 | **0.175** | 0.196 | **0.180** | 0.251 | 0.183 | **0.177** | 0.351 | 0.226 | 0.271 | +0.002 |
| | 336 | **0.318** | 0.358 | **0.333** | 0.403 | 0.390 | **0.331** | 0.424 | 0.367 | 0.460 | +0.013 |
| | 720 | **0.923** | **0.954** | **0.941** | 1.364 | 1.038 | 0.970 | 1.058 | 0.964 | 1.195 | +0.018 |

We first evaluate MixLinear with four benchmark LTSF datasets. Table 1 lists the MSE values of prediction accuracy under MixLinear and our baseline models at the forecast horizons of 96, 192, 336, and 720.

**Performance in Low-Channel Scenarios.** As Table 1 shows, MixLinear demonstrates strong performance in scenarios with fewer channels (7 channels), such as the ETTh1 and ETTh2 datasets. For instance, on ETTh1, MixLinear outperforms the baseline models, achieving the lowest MSE values of 0.351, 0.395, 0.411, and 0.423 at forecast horizons of 96, 192, 336, and 720, respectively. Specifically, MixLinear achieves an MSE reduction of 5.3% (+0.023) on ETTh1 at the forecast horizon of 336. On ETTh2, MixLinear ranks within the top two models across all horizons, except at the horizon of 96. These results highlight that our linear time-domain and frequency-domain decomposition method is well-suited for datasets with fewer channels.

**Performance in High-Channel Scenarios.** Table 1 further shows that MixLinear consistently delivers strong performance on datasets with a higher number of channels, such as Electricity (321 channels) and Traffic (862 channels). MixLinear ranks within the top two across most cases when compared with lightweight models like DLinear, FITS, and SparseTSF. Even when compared with parameter-heavy models, MixLinear still ranks within the top three in the majority of cases. Notably, MixLinear achieves this performance with only $0.1K$ parameters, significantly fewer than the baseline models, which require $6M$ parameters for PatchTST, $10K$ parameters for FITS, and $1K$ parameters for SparseTSF.

**Performance at Extended Horizons.** At the extended forecast horizon of 720, MixLinear consistently ranks within the top two models across all datasets, with the exception of the Electricity dataset (see Table 1). On ETTh1, MixLinear reduces the MSE by 0.003 at the 720 forecast horizon. On other datasets, the MSE increase at this horizon remains under 0.005. These findings highlight the robustness of MixLinear in handling long-term forecasting tasks effectively.

The experimental results with the dataset with ultra-long periods can be found in Appendix B.1.

## 4.3 EFFICIENCY

Table 2: Static and runtime metrics of MixLinear and the baselines on the Electricity dataset with a forecast horizon 720. The look-back length for each model is set to the default value used in those papers.

| Model | Parameters | MACs | Training Time(s) | Inference Time(ms) | MSE |
|---|---|---|---|---|---|
| Informer (2021) | 12.53M | 3.97G | 70.1 | 10.2 | 0.373 |
| Autoformer (2021) | 12.92M | 4.41G | 107.7 | 42.3 | 0.254 |
| FEDformer (2022) | 17.98M | 4.41G | 238.7 | 51.4 | 0.244 |
| FiLM (2022) | 12.22M | 4.41G | 78.3 | 36.1 | 0.236 |
| PatchTST (2023) | 6.31M | 11.21G | 290.4 | 108.1 | 0.210 |
| DLinear (2023) | 485.3K | 156M | 36.2 | 1.1 | 0.204 |
| FITS (2024) | 10.5K | 79.9M | 25.7 | 0.8 | 0.212 |
| SparseTSF (2024) | 0.92K | 12.71M | 33 | 0.9 | 0.205 |
| MixLinear (Ours) | 0.195K | 9.86M | 23.9 | 0.6 | 0.209 |

To examine the efficiency of MixLinear, we measure three static and runtime metrics including:

- **Parameters**: The number of parameters in the model. This is a measure of the model's complexity.

- **MACs**: The number of multiply-accumulate operations required per prediction. This is a measure of the model's computational cost.

- **Training Time(s)**: The amount of time (in seconds) it takes to train the model for one epoch. An epoch is one pass through the entire training dataset.

- **Inference Time(ms)**: Inference Time (ms): The amount of time (in milliseconds) it takes for the model to process input data and generate predictions during a forward pass.

Table 2 lists the measurements when we apply MixLinear and our baselines on the Electricity dataset with a forecast horizon of 720. As Table 2 lists, MixLinear is the most computationally efficient model among all solutions. To achieve a good prediction accuracy ($MSE = 0.209$), it only needs $0.195K$ parameters and $9.86M$ MACs, and provides the shortest training time of $23.9s$ per epoch. As a comparison, DLinear requires $485.3K$ parameters, $156M$ MACs, and a training time of $36.2s$ per epoch to achieve the best prediction accuracy ($MSE = 0.204$). The slight decreases in prediction accuracy are in exchange for a proportionally much larger enhancement in efficiency. We observe similar results in other datasets. The results show that MixLinear is well-suited for the scenarios where the computational resources are limited.

## 4.4 EFFECTIVENESS OF MIXING THE TIME AND FREQUENCY DOMAIN

Table 3: MSE results of multivariate LTSF with MixLinear when disabling part of the modules.

| Dataset | ETTh1 | | | | ETTh2 | | | | Electricity | | | | Traffic | | | |
|---|---|---|---|---|---|---|---|---|---|---|---|---|---|---|---|---|
| Horizon | 96 | 192 | 336 | 720 | 96 | 192 | 336 | 720 | 96 | 192 | 336 | 720 | 96 | 192 | 336 | 720 |
| TLinear | 0.376 | 0.398 | 0.412 | 0.425 | 0.317 | 0.366 | 0.369 | 0.389 | 0.181 | 0.192 | 0.209 | 0.245 | 0.485 | 0.483 | 0.520 | 0.528 |
| FLinear | 0.434 | 0.438 | 0.473 | 0.474 | 0.364 | 0.381 | 0.383 | 0.411 | 0.171 | 0.179 | 0.191 | 0.248 | 0.397 | 0.436 | 0.442 | 0.478 |
| MixLinear | **0.351** | **0.395** | **0.411** | **0.423** | **0.283** | **0.337** | **0.356** | **0.380** | **0.138** | **0.154** | **0.170** | **0.209** | **0.384** | **0.398** | **0.416** | **0.451** |

To evaluate the effectiveness of mixing time and frequency domain features, we compare MixLinear against two altered versions: TLinear and FLinear. TLinear is created by disabling the transformation in the frequency domain in MixLinear, while FLinear is implemented by disabling the transformation in the time domain. As Table 3 lists, TLinear achieves better performance on the ETTh1 and ETTh2 datasets compared to FLinear in the low-channel scenario. In the high-channel scenario, including the Electricity dataset with 321 channels and the Traffic dataset with 862 channels, FLinear tends to perform better. The reason behind is that the trend components of the time series data in the time domain are relatively easy to capture when there are a small number of channels because the model can focus on the long-term patterns in the individual time series. On the other hand, capturing the trend components becomes more effective in the frequency domain when facing a large number of variates, because the decomposition into different frequency bands benefits from the diversity of the channels. MixLinear outperforms both TLinear and FLinear in all cases because both Time Domain Transformation and Frequency Domain Transformation contribute significantly to the model's high forecasting accuracy.

## 4.5 GENERALIZATION ABILITY OF THE MIXLINEAR MODEL

Table 4: Comparison of generalization capabilities between MixLinear and other mainstream models. "Dataset A → Dataset B" denotes the training and validation on the training and validation sets of Dataset A, followed by testing on the test set of Dataset B.

| Dataset | ETTh2 → ETTh1 | | | | Electricity → ETTh1 | | | | Exchange → ETTh2 | | | | Solar → ETTh2 | | | |
|---|---|---|---|---|---|---|---|---|---|---|---|---|---|---|---|---|
| Horizon | 96 | 192 | 336 | 720 | 96 | 192 | 336 | 720 | 96 | 192 | 336 | 720 | 96 | 192 | 336 | 720 |
| Informer (2021) | 0.844 | 0.921 | 0.898 | 0.829 | \ | \ | \ | \ | \ | \ | \ | \ | \ | \ | \ | \ |
| Autoformer (2021) | 0.978 | 1.058 | 0.944 | 0.921 | \ | \ | \ | \ | \ | \ | \ | \ | \ | \ | \ | \ |
| FEDformer (2022) | 0.878 | 0.927 | 0.929 | 0.976 | \ | \ | \ | \ | \ | \ | \ | \ | \ | \ | \ | \ |
| FiLM (2022) | 0.876 | 0.904 | 0.919 | 0.925 | \ | \ | \ | \ | \ | \ | \ | \ | \ | \ | \ | \ |
| PatchTST (2023) | 0.449 | 0.478 | 0.426 | 0.400 | 0.424 | 0.475 | 0.472 | 0.470 | 0.459 | 0.573 | 0.617 | **0.556** | 0.503 | 0.537 | 0.446 | 0.536 |
| DLinear (2023) | 0.430 | 0.478 | 0.458 | 0.506 | 0.397 | 0.424 | 0.477 | 0.470 | 0.478 | 0.475 | 0.768 | 1.825 | 0.433 | 0.901 | 0.818 | 0.976 |
| FiTs (2024) | 0.419 | 0.427 | 0.428 | 0.445 | 0.380 | 0.414 | 0.440 | 0.448 | 0.444 | 0.532 | 0.581 | 0.600 | 0.371 | **0.374** | 0.398 | **0.419** |
| SparseTSF (2024) | 0.370 | 0.401 | 0.412 | **0.419** | **0.373** | **0.409** | 0.433 | 0.439 | 0.413 | 0.515 | 0.607 | 0.582 | **0.369** | 0.384 | 0.398 | 0.423 |
| MixLinear (Ours) | **0.361** | **0.388** | **0.406** | 0.427 | **0.373** | 0.410 | **0.428** | **0.434** | **0.406** | **0.507** | **0.542** | 0.600 | **0.369** | 0.388 | **0.397** | 0.422 |

MixLinear enhances forecasting ability by combining time and frequency domain features, improving generalization across datasets with similar periodicities. To explore this, we examined the cross-domain generalization performance of the MixLinear model by training on one dataset and testing on another. We compare MixLinear with several other mainstream models for multivariate LTSF in two scenarios: training and validation on ETTh2 with testing on ETTh1, and training and validation on Electricity with testing on ETTh1. As Table 4 lists, MixLinear has the best generalization ability and consistently achieves the lowest MSE values on different datasets and prediction horizons. When we train the model on ETTh2 and validate it on ETTh1, MixLinear achieves the lowest MSE

at forecast horizons of 96, 192, and 336. Similarly, when we train the model on Electricity and validate it on ETTh1, it achieves the lowest MSE at horizons of 96, 336, and 720, and offers an MSE of 0.410 at the 192 horizon which is just 0.001 higher than the best-performing model, SparseTSF (0.409). By combining features from both time and frequency domains, MixLinear can avoid the shortcut learning (Geirhos et al., 2020) problem, which occurs when the model focuses on time-space features while overlooking crucial underlying concepts in the frequency-space domain or vice versa, leading to limited poor performance on data unseen during training (He et al., 2023). The results highlight the effectiveness of MixLinear in transferring knowledge learned from one dataset to another by combining both time domain and frequency domain features, which demonstrates its robustness and adaptability in various forecasting scenarios.

## 5 RELATED WORK

### 5.1 LONG-TERM TIME SERIES FORECASTING

LTSF aims to predict future values over extended horizons, which is challenging because the time series data is complex and high-dimensional Zheng et al. (2024; 2023). The traditional statistical methods, such as ARIMA (Contreras et al., 2003) and Holt-Winters (Chatfield & Yar, 1988), are effective for short-term forecasting but often fall short for long-term predictions. Machine learning models, such as SVM (Wang & Hu, 2005), Random Forests Breiman (2001), and Gradient Boosting Machines (Natekin & Knoll, 2013), have improved performance by capturing non-linear relationships but require extensive feature engineering. Recently, deep learning models, such as RNNs, LSTMs, GRUs, and Transformer-based models including Informer and Autoformer have excelled in efficiently modeling long-term dependencies. The hybrid models that combine statistical and machine learning or deep learning techniques have also shown enhanced accuracy. State-of-the-art models like FEDformer (Zhou et al., 2022b), FiLM (Zhou et al., 2022a), PatchTST Nie et al. (2023), and SparseTSF incorporate advanced mechanisms like frequency domain transformations and efficient self-attention to achieve remarkable performance.

Recently, there has been a notable trend towards designing lightweight LTSF models. DLinear (Zeng et al., 2023) shows that even simple models can capture significant temporal periodic dependencies effectively. LightTS (Campos et al., 2023), TiDE (Das et al., 2023), and TSMixer (Chen et al., 2023) show similar conclusions. FITS (Xu et al., 2024) has emerged as a significant advancement in the field and achieved a milestone by scaling LTSF models to around $10K$ parameters while maintaining high predictive accuracy. FITS accomplishes this by transforming time-domain forecasting tasks into frequency-domain equivalents and employing low-pass filters to minimize parameter requirements. SparseTSF (Lin et al., 2024) pushes the boundaries even further by leveraging the Cross-Period Sparse Forecasting technique.

### 5.2 TIME SERIES DATA DECOMPOSITION

Several decomposition methods in the time domain have been introduced in the literature to handle such a task, including STL (Robert, 1990), TBATS (De Livera et al., 2011), and STR (Dokumentov et al., 2015) for periodic series, as well as $\ell_1$ trend filtering (Moghtaderi et al., 2011) and mixed trend filtering (Tibshirani, 2014) for non-periodic data. Although these techniques have gained popularity and proven effective in various applications, they exhibit limitations due to three reasons: the inefficiency in handling time series with long seasonal periods, the frequent seasonal shifts and fluctuations in real-world data, and the lack of robustness to outliers and noise (Gao et al., 2020).

Decomposing time series data in the frequency domain provides compressed representations that capture rich underlying patterns (Xu et al., 2020). These representations offer a more compact and efficient depiction of the inherent characteristics within the data (Xu et al., 2024). FEDformer decomposes sequences into multiple frequency domain modes using frequency transforms to extract features (Zhou et al., 2022b). TimesNet (Wu et al., 2022) employs a frequency-based method to separate intraperiod and interperiod variations. FITS (Xu et al., 2024) leverages this property by transforming the time series into the frequency domain, treating the data as a signal that can be expressed as a linear combination of sinusoidal components, a process that ensures no information loss. Each sinusoidal component is defined by its own frequency, amplitude, and initial phase, allowing for a precise representation of different oscillatory patterns present in the data. Although there

are many ways of frequency domain decomposition method, extracting features from the frequency domain requires suitable techniques. There will be many interferences in the signal, and suitable schemes for temporal features must be considered when combining deep learning methods.

# 6 CONCLUSION

There has been a growing interest in LTSF, which aims to predict long-term future values by identifying patterns and trends in large amounts of historical time-series data. A key challenge in LTSF is to manage long sequence inputs and outputs without incurring excessive computational or memory overhead, particularly in resource-constrained scenarios. In this paper, we introduce MixLinear, the first lightweight LTSF model that captures temporal and frequency features from both time and frequency domains. MixLinear applies the trend segmentation in the time domain to capture the intra-segment and inter-segment variations and captures the amplitude and phase information by reconstructing the trend spectrum from a low dimensional frequency domain latent space. Experimental results show that MixLinear can achieve comparable or better forecasting accuracy with only $0.1K$ parameters. Besides, MixLinear exhibits strong generalization capability and is well-suited for scenarios where the training data are limited.

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

# A  More Details of MixLinear

## A.1  Overview Workflow

---

**Algorithm 1:** Overall Pseudocode of MixLinear

---

**Input** : Historical look-back window $x_{t-L+1:t} \in \mathbb{R}^L$ and its period $w$
**Output:** Forecasted output $\hat{x}_{t+1:t+H} \in \mathbb{R}^H$

1: $x_{mean} \leftarrow \frac{1}{L} \sum_{i=t-L+1}^{t} x_i$  ▷ Compute the mean value of the historical look-back window
2: $x_{norm} \leftarrow x_{t-L+1:t} - x_{mean}$  ▷ Normalize the input by subtracting the mean
3: $x_{norm} \leftarrow \text{Conv1d}(x_{norm}, w) + x_{norm}$  ▷ Apply a 1D convolution over the normalized input sequence
4: $n \leftarrow \lceil L/w \rceil$  ▷ Determine the downsampled sequence length $n$
5: $x_{trend} \leftarrow \text{Reshape}(x, (n, w))$  ▷ Reshape the input into an $n \times w$ matrix for further processing
6: $\hat{n} \leftarrow \lceil \sqrt{\text{n}} \rceil^2$  ▷ Adjust the sequence length $\hat{n}$ to ensure $\sqrt{\hat{n}}$ is an integer
7: $x_{trend} \leftarrow \text{Pad}(x_{trend}, (\hat{n} - n))$  ▷ Apply zero-padding to extend the length $n$ to $\hat{n}$
8: $X_{Seg} \leftarrow \text{Reshape}(x_{trend}, (\sqrt{\hat{n}}, \sqrt{\hat{n}}))$  ▷ Reshape the trend data into a $\sqrt{\hat{n}} \times \sqrt{\hat{n}}$ matrix
9: $X_{Tp} \leftarrow \text{Linear}(\text{Linear}(X_{Seg})^T)^T$  ▷ Apply two linear transformations
10: $m \leftarrow \lceil H/w \rceil$  ▷ Compute the downsampled length of the forecast horizon $m$
11: $x_T \leftarrow \text{Reshape}(X_{Tp}, m)$  ▷ Reshape $X_{Tp}$ into a sequence of length $m$ for the forecast
12: $x_S \leftarrow \text{FFT}(x_{trend})$  ▷ Apply the FFT on the trend data with Equation 1
13: $x_S^{LPF} \leftarrow \text{LPF}(x_S, n^{LPF})$  ▷ Apply a low-pass filter to the frequency-domain representation to reduce noise
14: $z_S \leftarrow \text{Linear}(x_S^{LPF})$  ▷ Project the filtered frequency components into a latent space using a linear transformation
15: $x_{Sp} \leftarrow \text{Linear}(z_S)$  ▷ Apply a linear transformation to the latent frequency representation
16: $x_F \leftarrow \text{iFFT}(x_{Sp})$  ▷ Apply the iFFT to reconstruct the frequency domain signal with Equation 2
17: $x_M \leftarrow x_T + x_F + c_{mean}$  ▷ Combine the time-domain and frequency-domain components and add back the mean
18: $\hat{x}_{t+1:t+H} \leftarrow \text{Reshape}(x_M, H)$  ▷ Reshape the combined signal back into a sequence of length $H$ for the forecast output

---

The complete workflow of MixLinear is outlined in Algorithm 1, which takes a univariate historical look-back window $x_{t-L+1:t}$ as input and outputs the corresponding forecast results $\hat{x}_{t+1:t+H}$. Multivariate time series forecasting can be effectively achieved by integrating the CI strategy, i.e., modeling multiple channels using a model with shared parameters.

## A.2  Detailed Dataset Description

Table 5: Statistics of the datasets.

| Dataset | Traffic | Electricity | Solar | Weather | Exchange | ETTh1 | ETTh2 | ETTm1 | ETTm2 |
|---|---|---|---|---|---|---|---|---|---|
| Channels | 862 | 321 | 137 | 21 | 8 | 7 | 7 | 7 | 7 |
| Sampling Rate | 1 hour | 1 hour | 10 min | 10 min | 1 day | 1 hour | 1 hour | 15 min | 15 min |
| Total Timesteps | 17,544 | 26,304 | 52,560 | 52,696 | 7,588 | 17,420 | 17,420 | 69,680 | 69,680 |

The seven benchmark datasets used in our experiments are as follows:

(1) The ETT[2] dataset, sourced from Informer (Zhou et al., 2021), consists of data collected every 15 minutes between July 2016 and July 2018, including load and oil temperature readings. The

---

[2]https://github.com/zhouhaoyi/ETDataset

ETTh1 and ETTh2 subsets are sampled at 1-hour intervals, while ETTm1 and ETTm2 are sampled at 15-minute intervals.

(2) The Electricity[3] dataset contains the hourly electricity consumption of 321 customers, spanning the period from 2012 to 2014.

(3) The Traffic[4] dataset comprises hourly road occupancy rates, recorded by sensors placed on freeways in the San Francisco Bay Area. The data is provided by the California Department of Transportation.

(4) The Weather[5] dataset includes local climatological data collected from approximately 1,600 locations across the United States, spanning a period of four years (2010 to 2013). Data points are recorded at 1-hour intervals.

(5) The Solar-Energy[6] dataset records the solar power production from 137 PV plants in Alabama State, which are sampled every 10 minutes in 2016.

(6) The Exchange-Rate[7] dataset collects the daily exchange rates of 8 foreign countries from 1990 to 2016.

## A.3 DETAILED BASELINE MODEL DESCRIPTION

We briefly describe the baseline models we used in this paper:

(1) Informer (Zhou et al., 2021) is a Transformer-based model that employs self-attention distillation to highlight dominant attention by halving the input to cascading layers, enabling efficient handling of extremely long input sequences. The source code is available at https://github.com/zhouhaoyi/Informer2020.

(2) Autoformer (Wu et al., 2021) is a Transformer-based model that introduces the Auto-Correlation mechanism, leveraging the periodicity of time series to discover dependencies and aggregate representations at the sub-series level. The source code is available at https://github.com/thuml/Autoformer.

(3) Pyraformer (Liu et al., 2021) is a Transformer-based model that captures temporal dependencies of different ranges in a compact multi-resolution fashion. The source code is available at https://github.com/ant-research/Pyraformer.

(4) FEDformer (Zhou et al., 2022b) is a Transformer-based model proposing seasonal-trend decomposition and exploiting the sparsity of time series in the frequency domain. The source code is available at https://github.com/DAMO-DI-ML/ICML2022-FEDformer.

(5) TimesNet (Liu et al., 2023) is a CNN-based model with TimesBlock as a task-general backbone. It transforms 1D time series into 2D tensors to capture intraperiod and interperiod variations. The source code is available at https://github.com/thuml/TimesNet.

(6) SCINet (Liu et al., 2022) is a recursive downsample-convolve-interact architecture that uses multiple convolutional filters to extract distinct yet valuable temporal features from the downsampled sub-sequences or features. The source code is available at https://github.com/cure-lab/SCINet.

(7) iTransformer (Wu et al., 2022) is a Transformer based architecture that applies the attention and feed-forward network on the inverted dimensions. The source code is available at https://github.com/thuml/iTransformer.

(8) PatchTST (Nie et al., 2023) is a transformer-based model utilizing patching and CI technique. It also enables effective pre-training and transfer learning across datasets. The source code is available at https://github.com/yuqinie98/PatchTST.

---

[3]https://archive.ics.uci.edu/ml/datasets/ElectricityLoadDiagrams20112014
[4]http://pems.dot.ca.gov
[5]https://www.bgc-jena.mpg.de/wetter/
[6]http://www.nrel.gov/grid/solar-power-data.html
[7]https://github.com/laiguokun/multivariate-time-series-data

(9) DLinear (Zeng et al., 2023) is an MLP-based model with just one linear layer, which outperforms Transformer-based models in LTSF tasks. The source code is available at https://github.com/cure-lab/LTSF-Linear.

(10) FITS (Xu et al., 2024) is a linear model that manipulates time series data through interpolation in the complex frequency domain. The source code is available at https://github.com/VEWOXIC/FITS.

(11) SparseTSF (Lin et al., 2024) a novel, extremely lightweight model for LTSF, designed to address the challenges of modeling complex temporal dependencies over extended horizons with minimal computational resources. The source code is available at https://github.com/lss-1138/SparseTSF.

## A.4 DETAILED EXPERIMENTAL SETUP

We implement MixLinear in PyTorch (Paszke et al., 2019) and train it using the Adam optimizer (Diederik, 2015) for 30 epochs with early stopping based on a patience of 10 epochs. We follow the procedures outlined in FITS and Autoformer to split the dataset (Wu et al., 2021). Specifically, the ETT datasets are divided into training, validation, and test sets with a $6 : 2 : 2$ ratio. The other datasets are split with a $7 : 1 : 2$ ratio. Both our model and baselines use the same normalization method (i.e., StandardScaler).

MixLinear has minimal hyperparameters due to its simple design. The period $w$ is chosen based on the inherent cycle of the data (e.g., $w = 24$ for the ETTh1 dataset) or reduced when the dataset exhibits a longer cycle. The batch size is determined by the number of channels in each dataset. The batch size is set to 256 for the datasets with fewer than 100 channels (e.g., ETTh1). The batch size is set to 128 for the datasets with fewer than 300 channels (e.g., Electricity). Such a configuration maximizes the GPU parallelism while preventing any out-of-memory issues. In addition, given the small number of learnable parameters in MixLinear, we use a relatively large learning rate of 0.02 to accelerate training.

The baseline results reported in this paper come from the first version of the FITS paper, where FITS uses a uniform input length of 720. To ensure a fair comparison, we also use an input length of 720. The input lengths of other baseline models are set according to the values used in their original implementations.

## B MORE RESULTS AND ANALYSIS

In this section, we evaluate MixLinear with the ultra-long period datasets and examine the effect of the low-pass filter cutoff frequency thresholds on its performance.

### B.1 ULTRA-LONG PERIOD SCENARIO

Table 6: MSE results on the datasets with ultra-long periods.

| Dataset | ETTm1 | | | | ETTm2 | | | | Weather | | | |
|---|---|---|---|---|---|---|---|---|---|---|---|---|
| Horizon | 96 | 192 | 336 | 720 | 96 | 192 | 336 | 720 | 96 | 192 | 336 | 720 |
| Informer (2021) | 0.672 | 0.795 | 1.212 | 1.166 | 0.365 | 0.533 | 1.363 | 3.379 | 0.300 | 0.598 | 0.578 | 1.059 |
| Autoformer (2021) | 0.505 | 0.553 | 0.621 | 0.671 | 0.255 | 0.281 | 0.339 | 0.433 | 0.266 | 0.307 | 0.359 | 0.419 |
| Pyraformer (2022b) | 0.543 | 0.557 | 0.754 | 0.908 | 0.435 | 0.730 | 2.308 | 3.625 | 0.389 | 0.622 | 0.739 | 1.004 |
| FEDformer (2022b) | 0.379 | 0.426 | 0.445 | 0.543 | 0.203 | 0.269 | 0.325 | 0.421 | 0.217 | 0.276 | 0.339 | 0.403 |
| TimesNet (2023) | 0.338 | 0.374 | 0.410 | 0.478 | 0.187 | 0.249 | 0.321 | 0.408 | 0.172 | 0.219 | 0.280 | 0.365 |
| PatchTST (2023) | 0.293 | 0.333 | 0.369 | 0.416 | 0.166 | 0.223 | 0.274 | 0.362 | 0.149 | 0.194 | 0.245 | 0.314 |
| DLinear (2023) | 0.299 | 0.335 | 0.369 | 0.425 | 0.167 | 0.221 | 0.274 | 0.368 | 0.176 | 0.218 | 0.262 | 0.323 |
| FITS (2024) | 0.305 | 0.339 | 0.367 | 0.418 | 0.164 | 0.217 | 0.269 | 0.347 | 0.145 | 0.188 | 0.236 | 0.308 |
| SparseTSF (2024) | 0.314 | 0.343 | 0.369 | 0.418 | 0.165 | 0.218 | 0.272 | 0.350 | 0.172 | 0.215 | 0.260 | 0.318 |
| MixLinear (ours) | 0.332 | 0.353 | 0.385 | 0.437 | 0.170 | 0.223 | 0.275 | 0.360 | 0.172 | 0.212 | 0.257 | 0.324 |

To evaluate the prediction performance of MixLinear in ultra-long period forecasting scenarios, we conduct additional experiments using the ETTm1, ETTm2, and Weather datasets. Table 6 presents the MSE values achieved by MixLinear and the baseline models when applied to ultra-long period

forecasting tasks. As Table 6 shows, MixLinear demonstrates competitive performance across these datasets, with only $0.1K$ parameters, and even surpasses several transformer-based models that employ millions of parameters, including Informer, Autoformer, PyraFormer, and FEDFormer.

## B.2 EFFECT OF LOW PASS FILTER CUTOFF FREQUENCY ON PERFORMANCE

Table 7: MSE results of multivariate LTSF using MixLinear with different LPF.

| Dataset | ETTh1 | | | | ETTh2 | | | | Electricity | | | | Traffic | | | |
|---|---|---|---|---|---|---|---|---|---|---|---|---|---|---|---|---|
| Horizon | 96 | 192 | 336 | 720 | 96 | 192 | 336 | 720 | 96 | 192 | 336 | 720 | 96 | 192 | 336 | 720 |
| MixLinear ($n^{LPF}$=1) | 0.351 | 0.399 | 0.412 | 0.440 | 0.289 | 0.353 | 0.373 | 0.386 | 0.170 | 0.184 | 0.202 | 0.238 | 0.449 | 0.450 | 0.469 | 0.510 |
| MixLinear ($n^{LPF}$=5) | 0.356 | 0.395 | 0.412 | 0.426 | 0.290 | 0.349 | 0.360 | 0.389 | 0.151 | 0.163 | 0.179 | 0.243 | 0.407 | 0.422 | 0.430 | 0.473 |
| MixLinear ($n^{LPF}$=10) | 0.376 | 0.397 | 0.413 | 0.425 | 0.294 | 0.341 | 0.357 | 0.387 | 0.171 | 0.155 | 0.171 | 0.210 | 0.396 | 0.411 | 0.417 | 0.454 |
| MixLinear ($n^{LPF}$=14) | 0.358 | 0.396 | 0.414 | 0.426 | 0.284 | 0.338 | 0.361 | 0.381 | 0.140 | 0.154 | 0.171 | 0.209 | 0.391 | 0.408 | 0.417 | 0.455 |
| MixLinear ($n^{LPF}$=15) | 0.358 | 0.398 | 0.413 | 0.427 | 0.284 | 0.343 | 0.356 | 0.383 | 0.139 | 0.154 | 0.171 | 0.209 | 0.392 | 0.404 | 0.421 | 0.454 |
| MixLinear ($n^{LPF}$=16) | 0.361 | 0.398 | 0.413 | 0.423 | 0.284 | 0.339 | 0.356 | 0.381 | 0.140 | 0.154 | 0.171 | 0.209 | 0.392 | 0.404 | 0.417 | 0.451 |
| MixLinear ($n^{LPF}$=17) | 0.362 | 0.397 | 0.413 | 0.428 | 0.289 | 0.362 | 0.357 | 0.380 | 0.139 | 0.154 | 0.171 | 0.210 | 0.390 | 0.408 | 0.417 | 0.453 |
| MixLinear ($n^{LPF}$=18) | 0.357 | 0.397 | 0.414 | 0.424 | 0.285 | 0.340 | 0.357 | 0.388 | 0.139 | 0.154 | 0.171 | 0.209 | 0.390 | 0.403 | 0.416 | 0.452 |
| MixLinear ($n^{LPF}$=19) | 0.360 | 0.396 | 0.413 | 0.423 | 0.283 | 0.337 | 0.359 | 0.380 | 0.139 | 0.154 | 0.171 | 0.209 | 0.390 | 0.406 | 0.416 | 0.452 |

To evaluate the effect of the cutoff frequency used by the low-pass filter, we vary the LPF cutoff frequency threshold from 1 to 19 across the forecast horizons of 96, 192, 336, and 720 and measure MixLinear's prediction performance. As Table 7 lists, the prediction performance of MixLinear decreases significantly as the LPF threshold decreases from 5 to 1. To achieve a balance between performance and computational efficiency[8], a cutoff frequency of 5 is generally optimal for resource-constrained environments. However, the performance on the ETTh2 dataset is less sensitive to variations in the LPF cutoff frequency. The results indicate that while the LPF can help reduce the model complexity, a more adaptive filtering strategy may be required for LTSF tasks to maintain optimal performance.

Table 8: MSE results of MixLinear with varied hyperparameter $w$

| Dataset | Horizon | $w = 2$ | $w = 4$ | $w = 8$ | $w = 16$ | $w = 24$ | $w = 36$ |
|---|---|---|---|---|---|---|---|
| ETTh1 | 96 | 0.433 | 0.374 | 0.425 | 0.392 | **0.351** | 0.388 |
| | 192 | 0.450 | 0.402 | 0.409 | 0.421 | **0.395** | 0.411 |
| | 336 | 0.488 | 0.417 | 0.454 | 0.449 | **0.411** | 0.438 |
| | 720 | 0.466 | 0.436 | 0.437 | 0.432 | **0.423** | 0.437 |
| ETTh2 | 96 | 0.287 | 0.286 | 0.284 | 0.289 | **0.282** | 0.305 |
| | 192 | 0.351 | 0.358 | 0.341 | 0.360 | **0.336** | 0.348 |
| | 336 | 0.366 | 0.364 | 0.365 | 0.364 | **0.356** | 0.370 |
| | 720 | 0.390 | 0.383 | 0.383 | 0.389 | **0.380** | 0.391 |
| Electricity | 96 | 0.172 | 0.258 | 0.158 | 0.159 | **0.138** | 0.177 |
| | 192 | 0.197 | 0.268 | 0.174 | 0.179 | **0.154** | 0.184 |
| | 336 | 0.213 | 0.206 | 0.187 | 0.204 | **0.170** | 0.200 |
| | 720 | 0.242 | 0.216 | 0.230 | 0.230 | **0.209** | 0.235 |
| Traffic | 96 | 0.548 | 0.739 | 0.468 | 0.452 | **0.389** | 0.475 |
| | 192 | 0.568 | 0.700 | 0.496 | 0.513 | **0.403** | 0.432 |
| | 336 | 0.575 | 0.646 | 0.500 | 0.542 | **0.416** | 0.489 |
| | 720 | 0.583 | 0.481 | 0.559 | 0.524 | **0.452** | 0.527 |
| Solar | 96 | 0.207 | **0.205** | 0.215 | 0.209 | 0.211 | 0.218 |
| | 192 | 0.231 | 0.233 | 0.238 | 0.238 | **0.227** | 0.230 |
| | 336 | 0.250 | 0.254 | 0.258 | 0.251 | **0.240** | 0.242 |
| | 720 | 0.252 | 0.250 | 0.259 | 0.252 | **0.240** | 0.243 |
| Exchange | 96 | **0.087** | 0.088 | 0.091 | 0.095 | 0.101 | 0.101 |
| | 192 | 0.184 | **0.179** | 0.181 | 0.186 | 0.194 | 0.194 |
| | 336 | **0.320** | 0.327 | 0.328 | 0.333 | 0.342 | 0.342 |
| | 720 | **0.922** | 0.923 | 0.930 | 0.942 | 0.950 | 0.950 |

---

[8]A higher LPF threshold corresponds to a larger model size.

### B.3 Effect of Downsampling Period $w$

To evaluate the effect of the period w used for downsampling, we vary the hyperparameter $w$ from 2 to 36 across the forecast horizons of 96, 192, 336, and 720 and measure MixLinear's prediction performance.

Table 8 presents the MSE results of MixLinear for different values of the hyperparameter $w$ across six datasets (ETTh1, ETTh2, Electricity, Traffic, Solar, and Exchange). The hyperparameter $w$ represents the period used during downsampling, and the results are evaluated across forecasting horizons of 96, 192, 336, and 720.

For most datasets, $w = 24$ achieves the best performance, as shown by the bolded values, indicating its effectiveness in capturing periodicity. However, on the Exchange dataset, smaller values of $w$ (e.g., $w = 2$) yield better results, suggesting a different temporal structure where shorter periods are more suitable.

Shorter horizons (e.g., 96 and 192) generally lead to lower MSE scores, reflecting the relative ease of short-term predictions compared to longer horizons (e.g., 720). These results highlight the importance of selecting an appropriate $w$ value for optimal model performance.

### B.4 Generalization

Table 9: Comparison of generalization capabilities between MixLinear and other mainstream models. "Dataset A → Dataset B" denotes the training and validation on the training and validation sets of Dataset A, followed by testing on the test set of Dataset B.

| Dataset | Horizon | Metrics | MixLinear | SparseTSF | FiTS | DLinear | PatchTST |
|---|---|---|---|---|---|---|---|
| ETTh2 → ETTh1 | 96 | MSE | **0.361** | 0.370 | 0.419 | 0.430 | 0.449 |
| | 192 | MSE | **0.388** | 0.401 | 0.427 | 0.478 | 0.478 |
| | 336 | MSE | **0.406** | 0.412 | 0.428 | 0.458 | 0.426 |
| | 720 | MSE | 0.427 | **0.419** | 0.445 | 0.506 | 0.400 |
| Electricity → ETTh1 | 96 | MSE | 0.373 | **0.373** | 0.380 | 0.397 | 0.424 |
| | 192 | MSE | 0.410 | **0.409** | 0.414 | 0.424 | 0.475 |
| | 336 | MSE | **0.428** | 0.433 | 0.440 | 0.477 | 0.472 |
| | 720 | MSE | **0.434** | 0.439 | 0.448 | 0.470 | 0.470 |
| Exchange → ETTh2 | 96 | MSE | **0.406** | 0.413 | 0.444 | 0.478 | 0.459 |
| | 192 | MSE | **0.507** | 0.515 | 0.532 | 0.475 | 0.573 |
| | 336 | MSE | **0.542** | 0.607 | 0.581 | 0.768 | 0.617 |
| | 720 | MSE | 0.600 | 0.582 | 0.600 | 1.825 | **0.556** |
| Solar → ETTh2 | 96 | MSE | **0.369** | 0.369 | 0.371 | 0.433 | 0.503 |
| | 192 | MSE | 0.388 | 0.384 | 0.374 | 0.901 | 0.537 |
| | 336 | MSE | **0.397** | 0.398 | 0.398 | 0.818 | 0.446 |
| | 720 | MSE | 0.422 | 0.423 | **0.419** | 0.976 | 0.536 |
| ETTh1 → ETTh2 | 96 | MSE | 0.290 | 0.293 | **0.282** | 0.296 | 0.389 |
| | 192 | MSE | 0.352 | **0.353** | 0.349 | 0.378 | 0.365 |
| | 336 | MSE | 0.374 | **0.373** | 0.376 | 0.436 | 0.494 |
| | 720 | MSE | **0.388** | 0.390 | 0.390 | 0.627 | 0.444 |
| ETTh1 → Electricity | 96 | MSE | 0.326 | 0.326 | **0.309** | 0.325 | 0.560 |
| | 192 | MSE | 0.377 | 0.374 | **0.363** | 0.371 | 0.554 |
| | 336 | MSE | 0.390 | 0.392 | **0.383** | 0.427 | 0.711 |
| | 720 | MSE | 0.404 | 0.404 | **0.401** | 0.570 | 0.812 |

Table 9 compares the generalization capabilities of MixLinear with several mainstream models for long-term time series forecasting. The evaluation involves training and validating models on one dataset and testing them on another, with MSE as the performance metric. Lower MSE values indicate better performance.

MixLinear consistently achieves better performance across most dataset combinations and forecasting horizons. It performs particularly well for shorter horizons, such as 96 and 192, where it often records the lowest MSE. SparseTSF, while competitive in some cases, is generally outperformed by

MixLinear, especially on tasks involving significant domain shifts, such as *Exchange → ETTh2* and *Solar → ETTh2*.

FiTS and DLinear show moderate performance but consistently fall behind MixLinear, particularly for longer horizons like 336 and 720. PatchTST demonstrates strong results in a few cases but fails to match MixLinear's overall robustness.

A trend observed is the increase in MSE as the forecasting horizon lengthens, reflecting the growing difficulty of long-term predictions. Despite this, MixLinear maintains its advantage, showcasing its ability to generalize across datasets and handle challenging tasks effectively. The table underscores MixLinear's robustness and reliability, making it a standout model for long-term time series forecasting.

### B.5 ERROR BARS EVALUATION

Table 10: The error bars of MixLinear with 5 runs.

| Dataset | Horizon | Metrics | Seed 1 | Seed 2 | Seed 3 | Seed 4 | Seed 5 | Mean | Std. |
|---|---|---|---|---|---|---|---|---|---|
| ETTh1 | 96 | MSE | 0.351 | 0.377 | 0.390 | 0.358 | 0.375 | 0.370 | 0.016 |
| | 192 | MSE | 0.395 | 0.399 | 0.396 | 0.410 | 0.396 | 0.399 | 0.006 |
| | 336 | MSE | 0.411 | 0.416 | 0.412 | 0.412 | 0.426 | 0.415 | 0.006 |
| | 720 | MSE | 0.423 | 0.426 | 0.424 | 0.428 | 0.423 | 0.425 | 0.002 |
| ETTh2 | 96 | MSE | 0.283 | 0.283 | 0.285 | 0.293 | 0.283 | 0.285 | 0.004 |
| | 192 | MSE | 0.337 | 0.341 | 0.336 | 0.339 | 0.340 | 0.339 | 0.002 |
| | 336 | MSE | 0.356 | 0.360 | 0.355 | 0.357 | 0.359 | 0.357 | 0.002 |
| | 720 | MSE | 0.380 | 0.381 | 0.385 | 0.380 | 0.380 | 0.381 | 0.002 |
| Electricity | 96 | MSE | 0.138 | 0.138 | 0.139 | 0.169 | 0.139 | 0.145 | 0.014 |
| | 192 | MSE | 0.154 | 0.154 | 0.154 | 0.154 | 0.154 | 0.154 | 0.000 |
| | 336 | MSE | 0.170 | 0.171 | 0.170 | 0.172 | 0.171 | 0.171 | 0.001 |
| | 720 | MSE | 0.209 | 0.209 | 0.210 | 0.210 | 0.209 | 0.209 | 0.001 |
| Traffic | 96 | MSE | 0.390 | 0.390 | 0.389 | 0.390 | 0.390 | 0.390 | 0.000 |
| | 192 | MSE | 0.403 | 0.404 | 0.409 | 0.408 | 0.409 | 0.407 | 0.003 |
| | 336 | MSE | 0.416 | 0.421 | 0.418 | 0.420 | 0.423 | 0.420 | 0.003 |
| | 720 | MSE | 0.452 | 0.452 | 0.453 | 0.454 | 0.453 | 0.453 | 0.001 |
| Solar | 96 | MSE | 0.212 | 0.211 | 0.210 | 0.210 | 0.211 | 0.211 | 0.001 |
| | 192 | MSE | 0.227 | 0.227 | 0.233 | 0.228 | 0.226 | 0.228 | 0.003 |
| | 336 | MSE | 0.240 | 0.242 | 0.241 | 0.240 | 0.242 | 0.241 | 0.001 |
| | 720 | MSE | 0.240 | 0.241 | 0.244 | 0.241 | 0.241 | 0.241 | 0.002 |
| Exchange | 96 | MSE | 0.088 | 0.092 | 0.088 | 0.088 | 0.088 | 0.089 | 0.002 |
| | 192 | MSE | 0.179 | 0.179 | 0.179 | 0.179 | 0.175 | 0.178 | 0.002 |
| | 336 | MSE | 0.327 | 0.318 | 0.327 | 0.329 | 0.324 | 0.325 | 0.004 |
| | 720 | MSE | 0.923 | 0.932 | 0.925 | 0.928 | 0.925 | 0.927 | 0.004 |

To evaluate the stability, robustness, and predictive performance of the MixLinear model across various datasets and forecasting horizons, we conducted five independent runs with different random seeds. The table highlights the model's sensitivity to random initialization and its ability to deliver consistent performance by reporting the corresponding mean and standard deviation.

Table 10 presents the MSE results of MixLinear across six datasets (ETTh1, ETTh2, Electricity, Traffic, Solar, and Exchange) for four forecasting horizons (96, 192, 336, and 720 time steps). For each dataset and horizon, the table includes the MSE values obtained from the five runs, along with the mean and standard deviation (Std.). The mean represents the model's average performance, while the Std. quantifies the variability in predictions due to random initialization.

The results demonstrate that MixLinear consistently achieves low standard deviations across most datasets and horizons, indicating high stability and robustness. The model performs reliably across different random seeds, with only minor variations in specific cases. Predictive errors tend to increase slightly for longer forecasting horizons, reflecting the added complexity of long-term predictions. Datasets such as Electricity and Traffic exhibit particularly low variability, showcasing

the reliability of MixLinear on these datasets. Overall, the table underscores MixLinear's ability to deliver robust and stable predictions under diverse conditions.

## B.6    MORE DETAILED COMPUTATIONAL COMPLEXITY ANALYSIS

In this section, we analyze the computational complexity of MixLinear, a lightweight model that combines both time-domain and frequency-domain transformations to enhance efficiency in multivariate time series forecasting.

MixLinear captures both intra-segment and inter-segment variations by isolating channel-specific and periodic information from trend components in the time domain. This approach segments the trend data into smaller parts, allowing efficient handling of multivariate dependencies and reducing parameter complexity from $O(n^2)$ to $O(n)$ for input and output sequences of length $L$, with a known period $w$ and a subsequence length $n = \lceil \frac{L}{w} \rceil$.

To model long-term dependencies in the frequency domain, MixLinear maps the decoupled time series subsequences (trends) into a latent frequency space, where it reconstructs the trend spectrum. This process allows for effective frequency representation across segments, capturing critical periodic variations.

To further enhance efficiency, MixLinear applies an aggregation downsampling step using a 1D CNN with a kernel size of $w$, resulting in a computational complexity of $O(w)$. Following this, in the frequency domain transformation, FFT and iFFT operations are applied to the downsampled trend segments of length $n$, introducing an additional time complexity of $O(n \log n)$ and space complexity of $O(n)$ (Baxley & Zhou, 2006).

In summary, the total time complexity of MixLinear is $O(w) + O(n \log n) + O(n)$, which simplifies to $O(n \log n)$, the total space complexity of MixLinear is $O(w) + O(n) + O(n)$, which simplifies to $O(n)$.

## B.7    LOSS CONVERGENCE VISUALIZATION

To showcase the **loss convergence** of MixLinear and compare it with other models, we analyze their performance on the Exchange dataset under the input-720-predict-720 settings. Figure 3 illustrates how the training loss evolves across epochs for each model. This comparison highlights the efficiency and stability of MixLinear relative to other state-of-the-art models.

MixLinear and PatchTST demonstrate smooth and consistent loss convergence, indicating robust training dynamics and effective optimization. FITS converges rapidly within the initial epochs, achieving stable loss values quickly, which is advantageous for scenarios requiring faster training. In contrast, SparseTSF and DLinear exhibit noticeable oscillations during training, suggesting potential instability or sensitivity to hyperparameter settings. Meanwhile, iTransformer shows steady loss reduction but converges more slowly compared to MixLinear and PatchTST. These results emphasize the strengths of MixLinear in achieving stable and efficient training while also revealing differences in convergence behavior among various models.

## B.8    PREDICTION VISUALIZATION

To showcase the prediction performance of MixLinear and compare it with other models, we present visualizations of their prediction results. Figures 4 and Figures 5 display the prediction results on the Exchange dataset for different models under two settings: input-720-predict-96 (Figure 4) and input-720-predict-192 (Figure 5). In these figures, the blue lines represent the ground truth values, while the orange lines denote the model predictions.

In Figure 4, the models predict 96 future time steps based on 720 past time steps. MixLinear shows strong alignment with the ground truth, capturing short-term patterns effectively. SparseTSF performs reasonably well, though slight deviations are observed. FITS and PatchTST closely follow the ground truth, demonstrating robust short-term forecasting capabilities. DLinear and iTransformer, however, exhibit larger deviations, indicating less accuracy for short-term predictions.

In Figure 5, the models are tasked with predicting 192 future time steps using 720 past time steps. MixLinear continues to perform accurately with minimal deviations, proving its effectiveness for

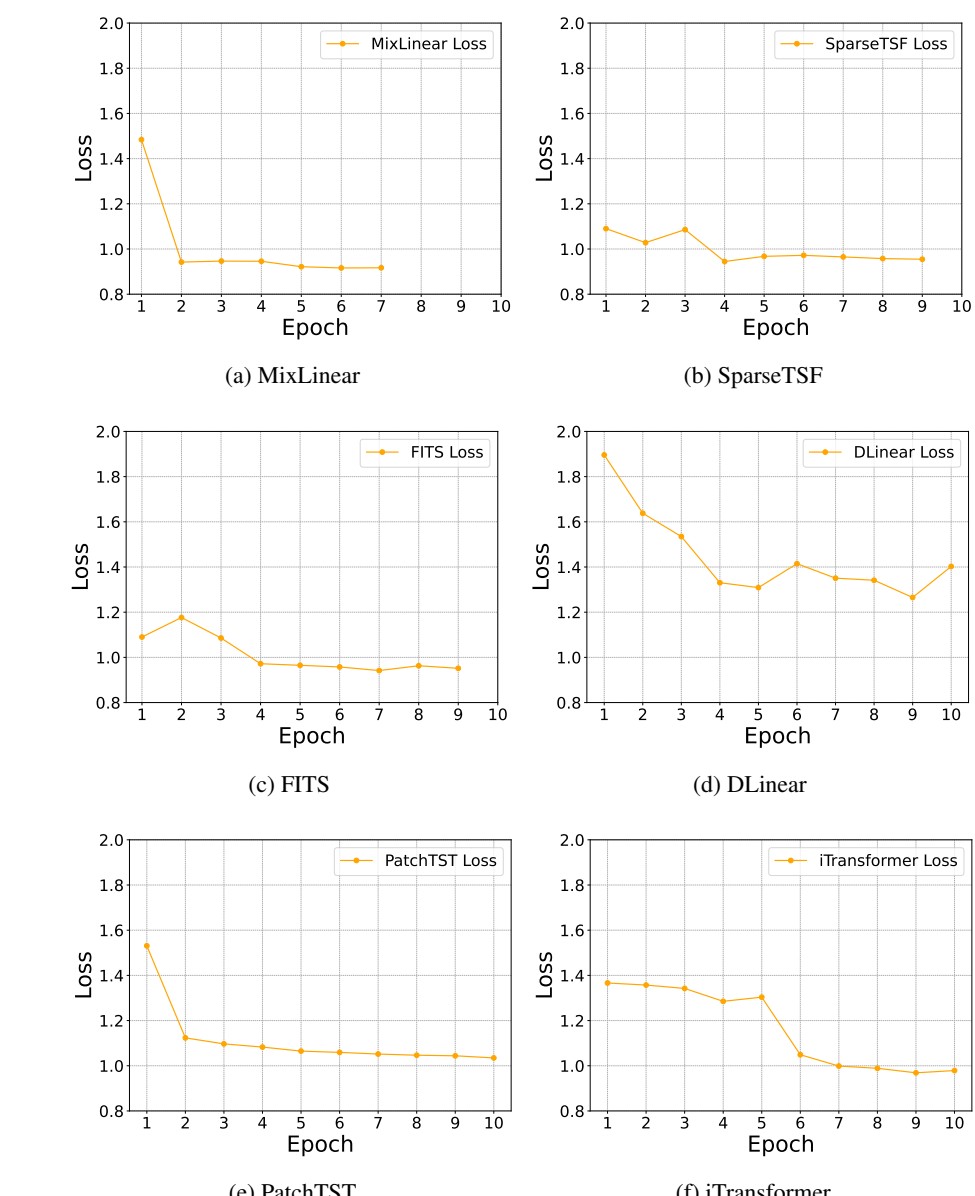

Figure 3: Loss Convergence on Dataset Exchange by different models under the input-720-predict-720 settings on Exchange dataset.

longer prediction horizons. SparseTSF and FITS display moderate accuracy but show occasional mismatches in trends. PatchTST maintains strong performance, similar to the 96-step setting, while DLinear and iTransformer show greater discrepancies and instability, struggling with the extended horizon.

Overall, these figures highlight the strengths and weaknesses of the models. MixLinear and PatchTST consistently deliver accurate predictions across both settings, whereas DLinear and iTransformer face challenges in capturing longer-term temporal patterns. This comparison underscores the importance of robust model design for both short-term and long-term forecasting.

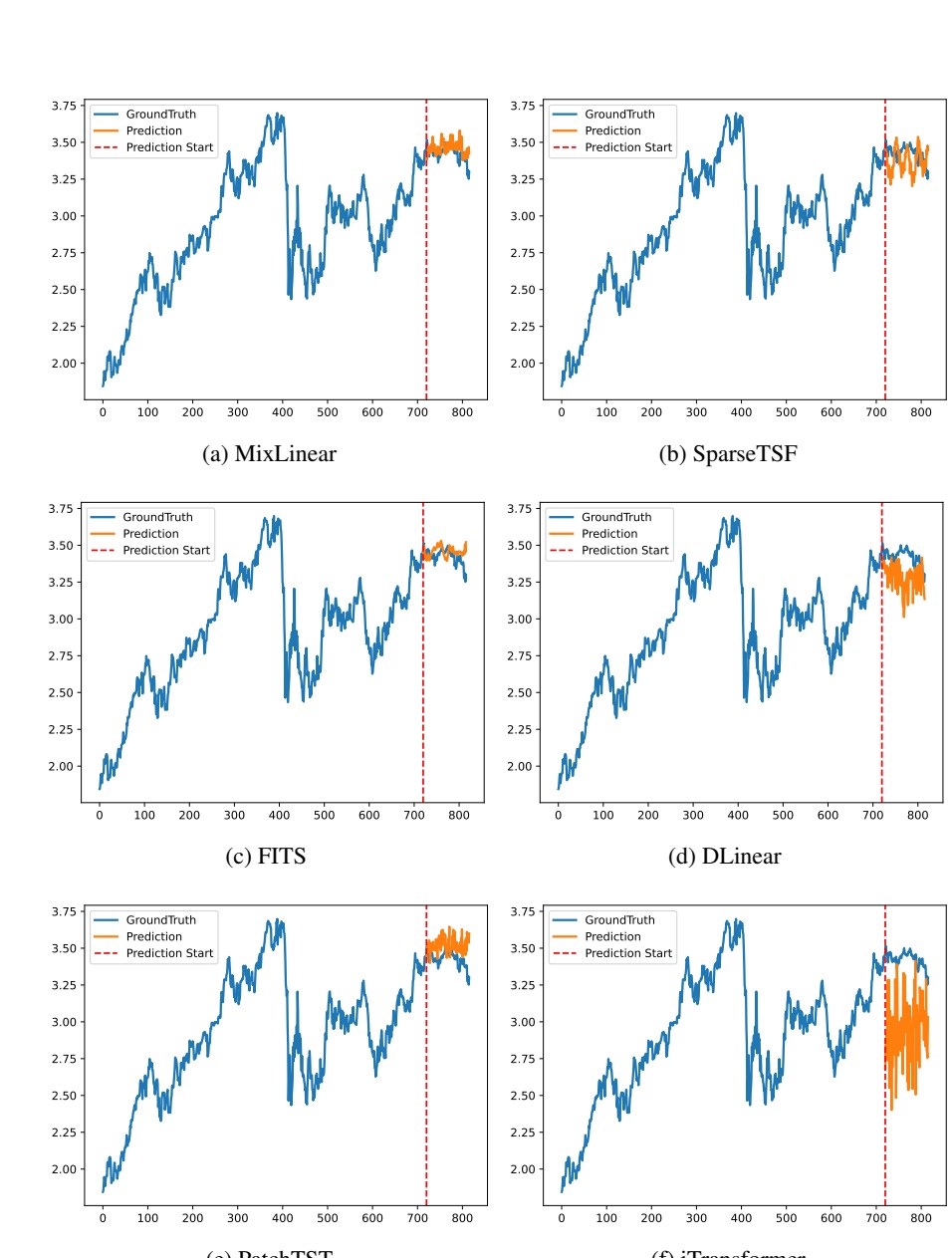

Figure 4: Prediction cases from Exchange by different models under the input-720-predict-96 settings. Blue lines are the ground truths and orange lines are the model predictions.

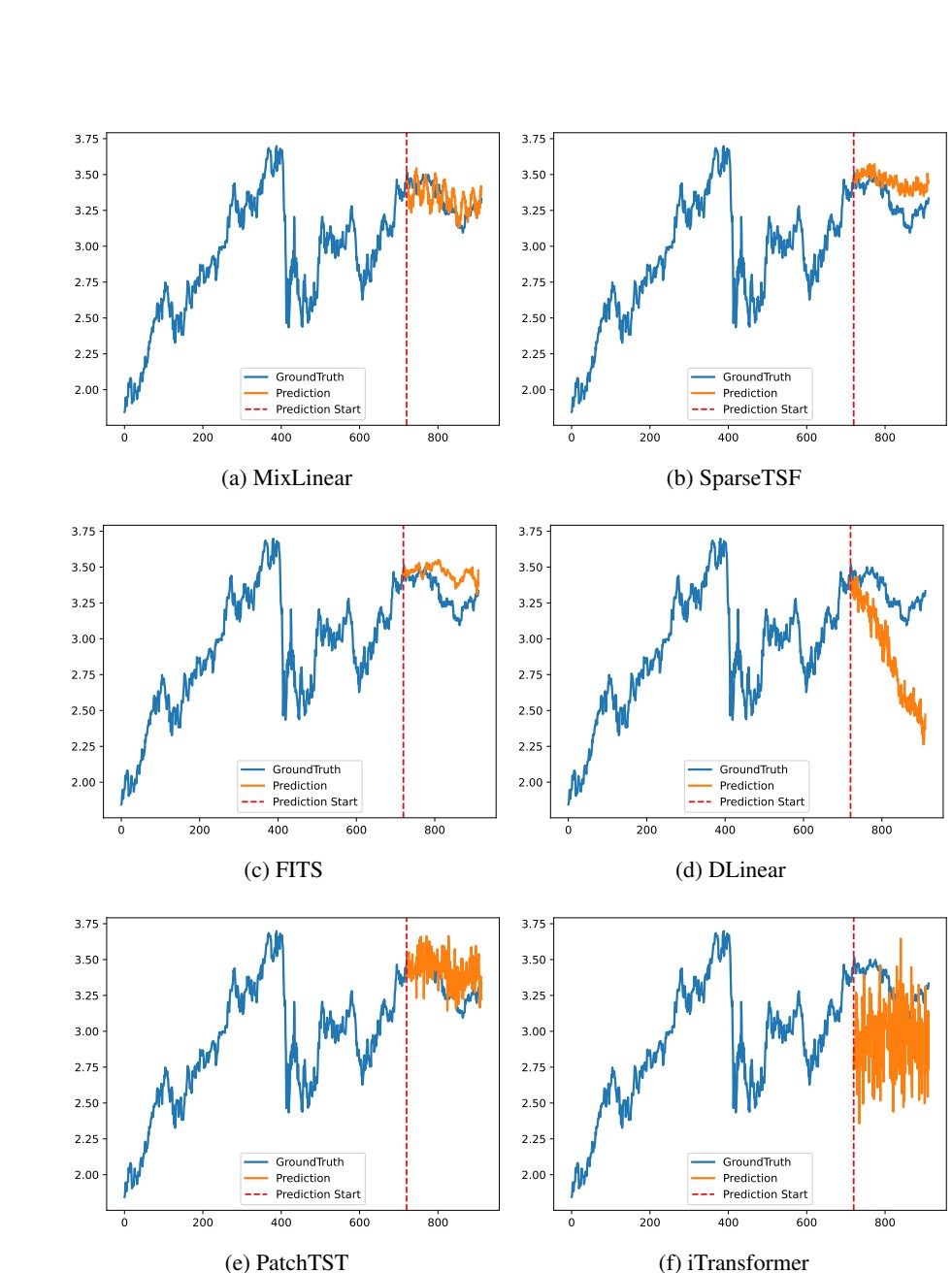

Figure 5: Prediction cases from Exchange by different models under the input-720-predict-192 settings. Blue lines are the ground truths and orange lines are the model predictions.

## B.9 PERFORMANCE COMPARISON BETWEEN MIXLINEAR AND NAIVE COMBINING FITS AND SPARSETSF

Table 11: MSE results of multivariate LTSF with MixLinear Compared with Simply Combining SparseTSF and FITS.

| Dataset | | ETTh1 | | | | ETTh2 | | | | Electricity | | | | Traffic | | | |
|---|---|---|---|---|---|---|---|---|---|---|---|---|---|---|---|---|---|
| FITS+SparseTSF | 10k | 0.384 | 0.419 | 0.448 | 0.440 | **0.271** | **0.332** | 0.360 | 0.388 | **0.133** | 0.158 | 0.174 | **0.203** | 0.402 | 0.413 | 0.428 | 0.449 |
| MixLinear | 0.1k | **0.351** | **0.395** | **0.411** | **0.423** | 0.283 | 0.337 | **0.356** | **0.380** | 0.138 | **0.154** | **0.170** | 0.209 | **0.384** | **0.398** | **0.416** | **0.451** |

To compare MixLinear with the naive combination of FITS and SparseTSF, we present the MSE results for multivariate LTSF in Table 11. As Table 11 presents, the results are reported across four datasets (ETTh1, ETTh2, Electricity, Traffic) for various input-output horizons.

The comparison reveals that MixLinear consistently outperforms the naive combination of SparseTSF and FITS in terms of lower MSE values across all datasets. For instance, in the ETTh1 dataset, MixLinear achieves a significantly lower MSE (0.351, 0.395, 0.411, and 0.423 for different horizons) compared to the combination model (0.384, 0.419, 0.448, and 0.440). Similar trends are observed in other datasets such as ETTh2 and Electricity, where MixLinear demonstrates better predictive accuracy.

The results highlight the superior design of MixLinear, which balances temporal and frequency information effectively. In contrast, the naive combination of SparseTSF and FITS struggles due to alignment and redundancy issues, leading to suboptimal performance. This table underscores the robustness and accuracy of MixLinear in multivariate LTSF tasks.

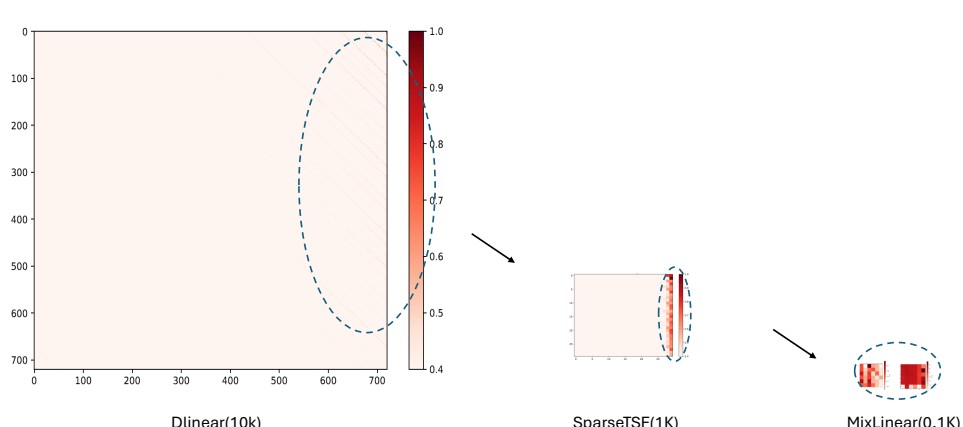

Figure 6: Model Weights Comparison in the Time Domain

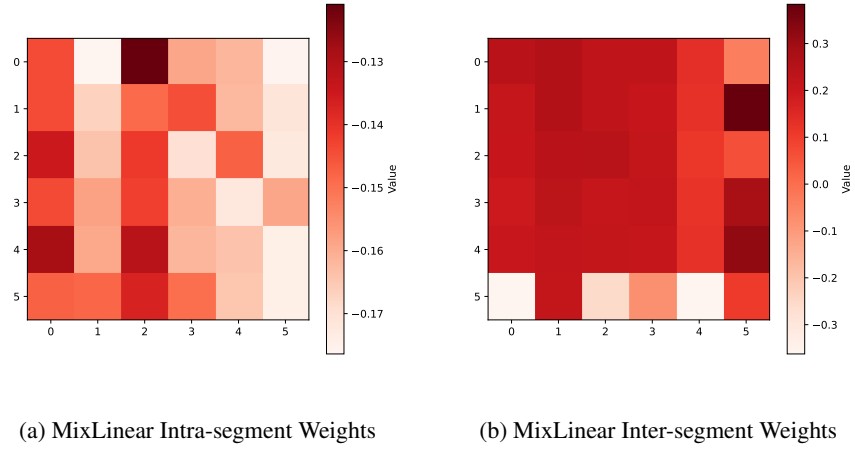

(a) MixLinear Intra-segment Weights      (b) MixLinear Inter-segment Weights

Figure 7: MixLinear Time Domain Weights

## C   TIME AND FREQUENCY DOMAIN VISUALIZATION

To showcase the patterns learned by the **Time Domain Transformation** and **Frequency Domain Transformation**, we visualize the weights learned by the MixLinear model. These visualizations offer valuable insights into how the model balances local and global feature extraction in the time domain while effectively managing compression and reconstruction in the frequency domain.

### C.1   MODEL WEIGHT COMPARISON IN THE TIME DOMAIN

Figure 6 showcase the time domain weights across three models trained on ETTh1 720-to-720 prediction scenario: DLinear, SparseTSF, and MixLinear. The progression showcases how the model's parameters are compressed while retaining significant temporal information.

DLinear (10k) is shown as the first panel, representing a model with 10,000 parameters. The heatmap displays the weight matrix, which covers the entire input sequence (over 720 steps). The evenly ditributed stripe on the right, highlighted with a dashed ellipse, indicates areas where significant weights are concentrated.

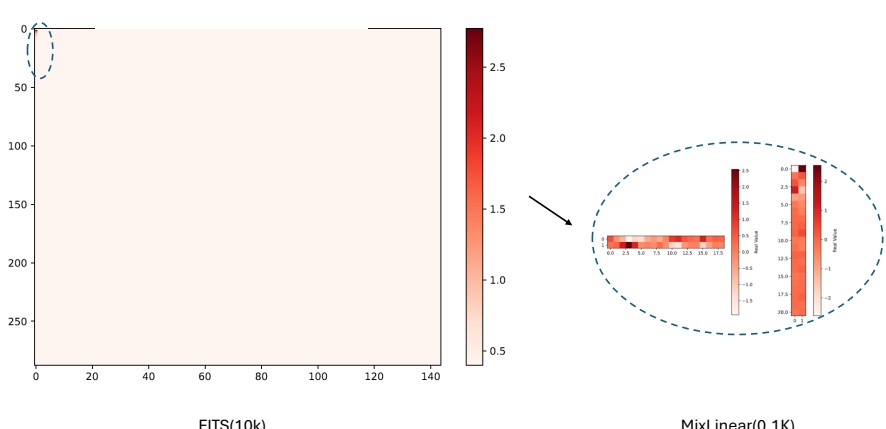

Figure 8: Model Weights Comparison in the Frequency Domain

SparseTSF (1K) is shown as the second panel, representing a more compact model with 1k parameters. Compared to DLinear, the weight matrix is significantly reduced in size. The key weight regions remain focused on a narrow band, suggesting targeted compression of influential features.

MixLinear (0.1K) is the final panel, representing an ultra-lightweight model with just 0.1k parameters. The weight matrix is further compressed, focusing on only a few crucial regions. The highlighted areas demonstrate how MixLinear preserves essential temporal patterns despite the extreme reduction in parameters.

This figure effectively illustrates the transition from large weight matrices (DLinear) to highly compressed, focused representations (MixLinear), emphasizing the efficiency and scalability of the proposed compression techniques for time-series forecasting models.

**Time Domain Weights**   Figure 7 illustrates the time domain weights of the MixLinear model, highlighting its two distinct components: intra-segment and inter-segment weights. Figure 7a shows the intra-segment weights, which capture the relationships and dependencies within individual segments of the input time series. The intensity of the colors reflects the magnitude of the weights, with darker regions indicating higher values. These weights focus on localized patterns, enabling the model to emphasize short-term relationships and trends effectively. Figure 7b depicts the inter-segment weights, which model the relationships and dependencies across different segments of the input time series. These weights help the model capture long-term patterns and interactions across segments, offering a holistic understanding of the temporal structure. Together, the intra- and inter-segment weights demonstrate the MixLinear model's ability to balance local and global feature extraction in the time domain. This visualization underscores the distinct yet complementary roles these components play in the model's architecture, enabling it to process both short-term and long-term temporal relationships effectively.

## C.2   MODEL WEIGHT COMPARISON IN THE FREQUENCY DOMAIN

Figure 8 presents a **Model Weights Comparison in the Frequency Domain** between FITS and MixLinear. This comparison emphasizes how MixLinear achieves an efficient representation of model weights in the frequency domain while significantly reducing the number of parameters.

The left panel depicts FITS, a model with 10k parameters. The heatmap illustrates a dense representation of weights across the frequency domain, with the most significant weights concentrated in the upper-left region (highlighted by the dashed ellipse). This dense matrix reflects the high parameter count and resource-intensive nature of FITS.

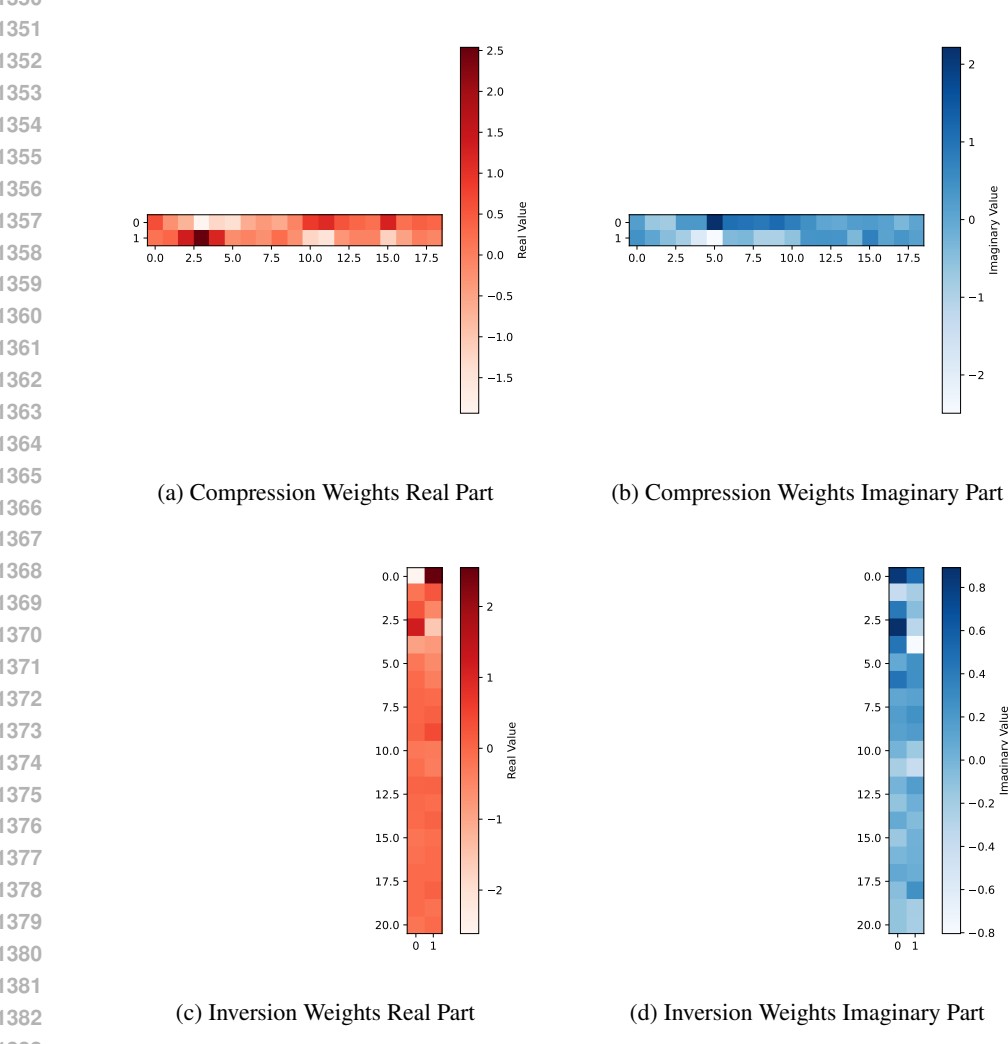

(a) Compression Weights Real Part

(b) Compression Weights Imaginary Part

(c) Inversion Weights Real Part

(d) Inversion Weights Imaginary Part

Figure 9: MixLinear Frequency Domain Weights

The right panel shows MixLinear, a model with only 100 parameters. In contrast to FITS, MixLinear exhibits a highly targeted weight distribution in the frequency domain. The dashed ellipse highlights how MixLinear focuses on a select few essential frequency components while discarding redundant or less significant weights. This approach enables MixLinear to achieve substantial compression and computational efficiency without compromising on capturing critical spectral features.

This figure highlights the effectiveness of MixLinear's lightweight design, demonstrating its ability to retain essential frequency-domain characteristics while operating with a fraction of the parameters required by FITS.

**Frequency Domain Weights** Figure 9 illustrates the frequency domain weights of the MixLinear model, which are divided into compression and inversion weights, each further split into real and imaginary components. Figure 9a presents the real part of the compression weights, which reflect how the model processes and compresses information in the frequency domain. Figure 9b shows the imaginary part of the compression weights, complementing the real part by encoding additional frequency-related patterns, such as phase and magnitude interactions. Figure 9c visualizes the real part of the inversion weights, which are responsible for reconstructing the data from the compressed frequency representation back into the time domain. Figure 9d illustrates the imaginary part of the inversion weights, which supports the reconstruction process by accounting for complex interactions in the frequency domain. These visualizations provide insight into how MixLinear handles trans-

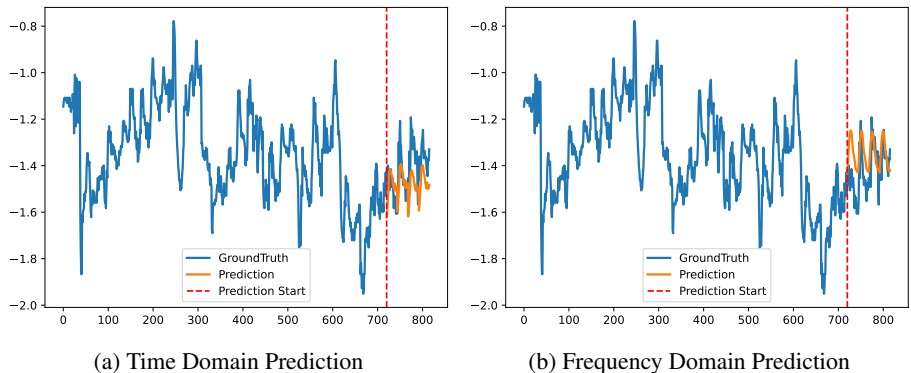

(a) Time Domain Prediction       (b) Frequency Domain Prediction

Figure 10: Comparison of MixLinear's Time Domain and Frequency Domain Modules on ETTh1 Fragment 1

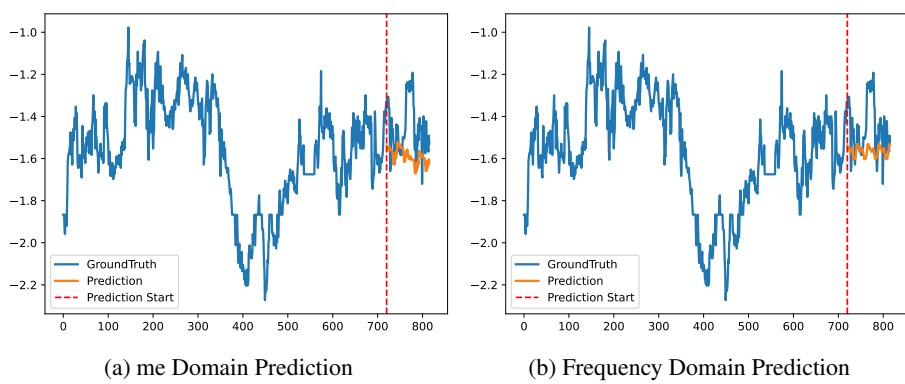

(a) me Domain Prediction       (b) Frequency Domain Prediction

Figure 11: Comparison of MixLinear's Time Domain and Frequency Domain Modules on ETTh1 Fragment 2

formations in the frequency domain. By effectively balancing compression and reconstruction and leveraging both real and imaginary components, the model ensures the integrity of the time series data while capturing critical patterns and features. This design highlights MixLinear's strength in processing complex temporal data and its adaptability to diverse time series structures.

### C.3 PREDICTION COMPARISON ACROSS TWO DOMAINS

Figure 10 and Figure 11 present a comparison of MixLinear's Time Domain and Frequency Domain modules on two fragments of data from ETTh1. As illustrated in these figures, the frequency domain predictions appear smoother due to the application of a low-pass filter, which eliminates high-frequency noise and allows the model to focus more on global trends. In contrast, the time domain predictions capture finer details, emphasizing localized variations in the data.

We have mathematically proven that combining time domain and frequency domain models reduces the uncertainty of future predictions more effectively than using either model alone. From an information theory standpoint, the combined model minimizes the conditional entropy and maximizes the mutual information with future observations. This superiority arises from the model's ability to capture a broader range of data structures, including trends, cycles, and other complex patterns inherent in time series data.

