# OpenReview forum: "MixLinear: Extreme Low Resource Multivariate Time Series Forecasting with $0.1k$ Parameters"
_ICLR.cc/2025/Conference — Submitted to ICLR 2025_

### Official Review · Reviewer_jXYM · 2024-10-16

**Soundness:** 3
**Presentation:** 2
**Contribution:** 2
**Rating:** 3
**Confidence:** 4

**Summary:**

The authors present an approach coined MixLinear to improve predictive performance for (multi-variate) time series problems with a long forecasting horizon. They put an emphasis on drastically reducing the necessary parameter count of their model compared to other existing approaches while maintaining or even improving on forecasting quality. For this, their proposed method decomposes the time signal in a number of steps into seasonal periods, trends as well as its frequency spectra. Through a learned mixing the information is combined to produce the forecast. Therein, information pieces are already shorter or subsampled to theoretically achieve near-linear complexity with respect to the number of time steps L in the historic window of a model's input sequence. In an experimental evaluation on well-known datasets like ETTh and Traffic, the proposed method is compared to other established Transformer-based forecasting approaches.

**Strengths:**

- The author touch an important topic of avoiding drastic overparametrization of (transformer-based) time series forecasting models

- An extensive comparison with other, current baseline models is performed

**Weaknesses:**

- You claim that "Transformer-based models offer high forecasting accuracy, they are often too compute-intensive to be deployed on devices with hardware constraints [...]". The reviewer feels that this claim is not substantiated in light of various pruning techniques (removing upto 99% of weights after training), knowledge destillation or neuron transplantation. Could you provide according proof? Your paper currently only considers the training phase, but not its later use in an inference setup.

- The manuscript claims "The high computational demands and large memory requirements of these models hinder their deployment for LTSF tasks on resource-constrained devices" - the authors do report in Section 4.3 on the efficiency of these models, but it is unclear to the reviewer how much effort went into optimizing these. If only MixLinear has been highly optimized and the others taken as is, the empirical evaluation has a consistent bias in favor of MixLinear.

- Carefully check the paper for grammatical erros as well as precise phrasing, examples include, but are not limited to
	* "On the other hand, the linear models aim [...]" (abstract) - the second 'the' is superfluous
	* "[...] overhead by employing either decomposition methods [...]" (abstract) - what does 'either' refer to here?
	* "[...] by modeling intra-segment and inter-segment variations [...]" (abstract) - segment is unclear, could you clarify this term?

- Fig. 1 shows an example of the Electricity dataset, which a) has not been introduced at that point and b) does not indicate if and how data has been normalized, making the MSE values hard to estimate. The same applies to the tables 1, 2 and 3 later in the manuscript's body. Could the authors improve the textual description?

- "[...] by decoupling channel and periodic information from the trend components [...]" - there are several prior works that attempt do achieve this. This includes, but is not limited to the following:
	* A. Weyrauch, et al., "ReCycle: Fast and Efficient Long Time Series Forecasting with Residual Cyclic Transformers," in 2024 IEEE Conference on Artificial Intelligence (CAI), Singapore, Singapore, 2024 pp. 1187-1194.
	* Heidrich, B., Turowski, M., Ludwig, N., Mikut, R., & Hagenmeyer, V. (2020, June). Forecasting energy time series with profile neural networks. In Proceedings of the eleventh acm international conference on future energy systems (pp. 220-230).
	* Pathak, J., Subramanian, S., Harrington, P., Raja, S., Chattopadhyay, A., Mardani, M., ... & Anandkumar, A. (2022). Fourcastnet: A global data-driven high-resolution weather model using adaptive fourier neural operators. arXiv preprint arXiv:2202.11214.

	How does your approach compare to these or other works? Especially drastically reduces computational complexity by computing mostly in the frequency domain, achieving a time complexity of O(L log L), where L is the sequence length

- The authors are kindly requested to improve Fig. 2 by
	* using the correct mathematical symbols, e.g. for the set of real numbers,
	* increasing font size, e.g. on the trend spectrum,
	* use full words without underscores,
	* define all mathematical symbols, e.g. LPF, in the caption,
	* have a consistent direction of reading (currently a mix between bottom-top and left-right).

- Can the authors comment on how the frequency domain part of MixLinear differs from Fourier Neural Operators?

	Li, Z., Kovachki, N., Azizzadenesheli, K., Liu, B., Bhattacharya, K., Stuart, A., & Anandkumar, A. (2020). Fourier neural operator for parametric partial differential equations. arXiv preprint arXiv:2010.08895.

- Is it meaningful to consistently use the Landau-/Big-O notation when assessing the algorithm's complexities, e.g. p. 4 in the subsection "Segment Transformation"

- Have the authors considered to only work on the differences between time steps or the residuals of the time series and a simple smoothing average curve? A vast number of publications have identified this to be beneficial for predictive performance, training convergance as well as a (further) reduced parameter count.

- It seems meaningful that the sign of "Improvement" in Table 1 should be inverted due to the MSE being lower indicating a better model. Furthermore, since it is not consistently an improvement, does the term "Difference to best baseline", or similar, make sense?

- The title and manuscript body speaks of 0.1K parameter models, whereas it is closer to 0.2K - it seems to the reviewer that it would not take away from the paper's message by being more precise.

- In its current form the paper only reports optimal forecasting but does not consider a distribution across multiple seeds. Is MixLinear consistent in achieving high predictive performance? How large is the spread compared to other approaches?

- The work is motivated by attempting to find models that work in resource constrained environments. Can you make an example for a possible usage scenario? Would it be used to train the model, make predictions or both? In its current form the manuscript puts a lot of emphasis on evaluating the training phase in an environment with a lot of compute power (A100-80). How would your approach actually fair in your envisioned usage scenario?

- In the manuscript only time series forecasting problems with strong seasonality are considered. It seems to the reviewer that the approach would not work well for problems where this is not applicable. It seems meaningful to clearly spell out this assumption.

**Questions:**

- Can the authors comment on the suggested method's capability to capture peaks or other extremes of the time series well?

- Can you comment on how MixLinear's hyperparameters can be effectively found, e.g. cutoff values?

- Are you willing to release the source code of your method and its evaluation for the sake of reproducibility?

---

> ### Author Response · Authors · 2024-11-22
>
> **Comment1:**
> In its current form, the paper only reports optimal forecasting but does not consider a distribution across multiple seeds. Is MixLinear consistent in achieving high predictive performance? How large is the spread compared to other approaches?
>
> **Response:**
> Thank you for your comment. To evaluate the stability, robustness, and predictive performance of the MixLinear model across various datasets and forecasting horizons, we conducted five independent runs with different random seeds. The table highlights the model's sensitivity to random initialization and its ability to deliver consistent performance by reporting the corresponding mean and standard deviation.
>
> We have added the new results into Appendix B.5 Table 10 Page 18 of the revised manuscript.
>
> ---
>
>
> **Comment2:**
> Can you comment on how MixLinear's hyperparameters can be effectively found, e.g., cutoff values?
>
> **Response:**
> To evaluate the effect of the cutoff frequency used by the low-pass filter, we conducted new experiments by varying the LPF cutoff frequency threshold from $ n^{\text{LPF}} = 1 $ to $ n^{\text{LPF}} = 19 $ and measured MixLinear's prediction performance. The results show that the prediction performance of MixLinear decreases significantly as the LPF threshold decreases from $ n^{\text{LPF}} = 5 $ to $ n^{\text{LPF}} = 1 $.
>
> To balance performance and computational efficiency, a cutoff frequency of $ n^{\text{LPF}} = 5 $ is generally optimal for resource-constrained environments.
>
> We have added these results to Appendix B.2, Table 7 (Page 16) of the revised manuscript.
>
> ---
>
>
> **Comment3:**
> The manuscript only considers time series forecasting problems with strong seasonality. It seems the approach would not work well for problems where this is not applicable. It seems meaningful to clearly spell out this assumption.
>
> **Response:**
> Thank you for your comment. We have added the Exchange dataset, which does not exhibit strong seasonality. Despite the lack of seasonal patterns, MixLinear demonstrates better predictive performance on this dataset, showcasing its versatility across diverse time series forecasting scenarios.
>
> Additionally, we have added evaluations on the Solar dataset to further assess MixLinear's performance. The new results have been included in Section 4.2, Table 1 (Page 6) of the revised manuscript.
>
> ---
>
>
> **Comment4:**
> You claim that "Transformer-based models offer high forecasting accuracy, but they are often too compute-intensive to be deployed on devices with hardware constraints [...]". This claim is not substantiated in light of various pruning techniques, knowledge distillation, or neuron transplantation. Could you provide proof? Your paper currently only considers the training phase, not the inference setup.
>
> **Response:**
> Thank you for your comment. We have presented the inference time for MixLinear and other models. The results show that MixLinear achieves superior performance while being the most lightweight model, with the least inference time among all methods evaluated.
>
> These findings highlight MixLinear's suitability for deployment on resource-constrained devices, even when compared to Transformer-based models optimized through techniques like pruning or distillation. We have added these results to Section 4.3, Table 2 (Page 7) of the revised manuscript.
>
> ---
>
>
> **Comment5:**
> "By decoupling channel and periodic information from the trend components [...]" - there are several prior works that attempt to achieve this. How does your approach compare to these or other works? Especially with regard to drastically reducing computational complexity by computing mostly in the frequency domain, achieving a time complexity of $ O(L \log L) $, where $ L $ is the sequence length.
>
> **Response:**
> Thank you for your comment and for highlighting these related works.
>
> - **ReCycle** employs primary cycle compression (similar to SparseTSF) to reduce the computational complexity of attention mechanisms in long time series. Our approach operates on the downsampled trend using linear layers and further enhances efficiency by leveraging both the time and frequency domains. This results in MixLinear (0.195K parameters) being significantly more lightweight than SparseTSF (0.92K parameters).
>
> - **Profile Neural Networks** utilize convolutional layers to model time series. In contrast, our method relies on simple linear layers, which are computationally efficient while capturing both temporal and spectral patterns.
>
> - **FourCastNet** applies Fourier transformations with a Vision Transformer backbone. Unlike FourCastNet, MixLinear employs linear layers in both domains to achieve superior efficiency with a time complexity of $ O(n \log n) $, where $ n = L / w $ and $ w $ is the period of the sequence.
>
> We have added this analysis to Appendix B.6 (Page 19) of the revised manuscript.
>
> ---

---

> ### Author Response · Authors · 2024-11-22
>
> **Comment6:**
> The manuscript claims, "The high computational demands and large memory requirements of these models hinder their deployment for LTSF tasks on resource-constrained devices". However, it is unclear how much effort went into optimizing these models. If only MixLinear has been highly optimized, the empirical evaluation may favor MixLinear.
>
> **Response:**
> Thank you for your comment. To ensure a fair comparison, we adopted the optimized hyperparameters provided in the official GitHub repositories of all baseline models. All datasets used are benchmark datasets, and baseline models have been optimized by their respective developers on these datasets.
>
> ---
>
>
> **Comment7:**
> Carefully check the paper for grammatical errors as well as precise phrasing. Examples include:
> - "On the other hand, the linear models aim [...]" - the second "the" is superfluous.
> - "Overhead by employing either decomposition methods [...]" - what does "either" refer to here?
> - "[...] by modeling intra-segment and inter-segment variations [...]" - "segment" is unclear. Could you clarify this term?
>
> **Response:**
> Thank you for your feedback. We have addressed these issues in the Abstract of the revised manuscript. "Segment" refers to a fragment of the time series data.
>
> ---
>
>
> **Comment8:**
> Fig. 1 shows an example of the Electricity dataset, but:
> 1. The dataset has not been introduced at this point.
> 2. It does not indicate if and how data has been normalized, making MSE values hard to estimate.
>
> **Response:**
> Thank you for your comment. We have introduced the Electricity dataset before the figure in the Introduction section. Regarding normalization, all models (including MixLinear and baselines) use the same normalization method (StandardScaler). This ensures consistency across comparisons. Details on normalization have been added to Appendix A.4 (Page 15) of the revised manuscript.
>
> ---
>
>
> **Comment9:**
> Please improve Fig. 2 by:
> - Using correct mathematical symbols, e.g., for the set of real numbers.
> - Increasing font size.
> - Defining all mathematical symbols (e.g., LPF) in the caption.
> - Ensuring consistent reading direction.
>
> **Response:**
> Thank you for your feedback. We have revised Figure 2 (Page 4) of the revised manuscript to address these issues.
>
> ---
>
>
> **Comment10:**
> How does the frequency domain part of MixLinear differ from Fourier Neural Operators (FNO)?
>
> **Response:**
> Thank you for your comment. FNO primarily focuses on learning resolution-invariant solution operators for parametric PDEs, such as the Navier-Stokes equations. In contrast, MixLinear leverages the frequency domain to capture spectral features like phase and amplitude patterns, specifically for time series forecasting. Unlike FNO, which is tailored for PDEs, MixLinear integrates spectral analysis to model temporal dependencies effectively while remaining computationally efficient.
>
> ---

---

> ### Author Response · Authors · 2024-11-22
>
> **Comment11:**
> Is it meaningful to consistently use the Landau-/Big-O notation when assessing the algorithm's complexities, e.g., p. 4 in the subsection "Segment Transformation"?
>
> **Response:**
> Thank you for your comment. We appreciate your suggestion regarding the consistent use of Landau/Big-O notation for assessing algorithmic complexities. We continue to use Landau/Big-O notation to ensure that all complexity analyses are consistently expressed for clarity and precision. This standardizes the discussion and provides a clear comparison of computational efficiency across different methods.
>
> ---
>
>
> **Comment12:**
> Have the authors considered working on the differences between time steps or the residuals of the time series and a simple smoothing average curve? A vast number of publications have identified this to be beneficial for predictive performance, training convergence, and reduced parameter count.
>
> **Response:**
> Thank you for your comment. We use 1D CNN aggregation to smooth the curve and incorporate the residual of the original series to preserve information while generating the trend. We have added discussions to Appendix A.1 and Algorithm 1 (lines 1-3) on Page 13 of the revised manuscript to clarify this.
>
> ---
>
>
> **Comment13:**
> It seems meaningful that the sign of "Improvement" in Table 1 should be inverted since lower MSE indicates a better model. Furthermore, since it is not consistently an improvement, does the term "Difference to best baseline" or similar make more sense?
>
> **Response:**
> Thank you for your suggestion. We have changed the term "Improvement" in Table 1 to "Difference." This change better reflects the nature of the metric, particularly considering that a lower MSE signifies superior performance.
>
> ---
>
>
> **Comment14:**
> Can the authors comment on the suggested method's capability to capture peaks or other extremes of the time series well?
>
> **Response:**
> Thank you for your comment. MixLinear is designed to effectively capture peaks and extremes in time series data by leveraging its dual-domain approach:
> - **Time Domain:** The model captures localized variations and dependencies through intra-segment and inter-segment transformations, enabling it to respond to abrupt changes or spikes.
> - **Frequency Domain:** MixLinear uses spectral analysis to identify and reconstruct critical patterns, including those associated with peaks and extremes.
>
> This complementary integration of temporal and frequency domain features ensures that MixLinear can model both subtle trends and sharp variations in the data.
>
> ---
>
>
> **Comment15:**
> Are you willing to release the source code of your method and its evaluation for the sake of reproducibility?
>
> **Response:**
> Thank you for your comment. We are committed to ensuring the reproducibility of our research and will release the source code along with the evaluation details to facilitate further exploration and validation by the community.

---

> > ### Comment · Reviewer_jXYM · 2024-11-25
> > **Changes appreciated, grade remains unchanged**
> >
> > The reviewer thanks the authors greatly for the extensive answer to the questions. While some of the answer are satisfactory and have improved the line of argument (e.g. the multiple seeds), several points remain open or vague (e.g. confusion about an unlearned smoothing average and the author's suggest 1D convolutions). Hence, the reviewer feels that the manuscript does not yet fulfill the quality criteria of this conference or would need to be so drastically changes that it requires and additional review phase. The initial scores therefore remains as stated.

---

> > > ### Author Response · Authors · 2024-11-25
> > >
> > > Dear Reviewer,
> > >
> > > As far as we know, our model, MixLinear, is the LTSF model with the smallest parameter count (0.1K) by far that achieves comparable predictive performance with state-of-the-art models such as SparseTSF (1K, ICML 2024 Oral), FITS (10K, ICLR 2024 Spotlight), and PatchTST (6.1M, ICLR 2024). The lightweight design of MixLinear makes it uniquely suitable for deployment on resource-constrained devices, such as Arm® Cortex®-M3 microcontrollers, for power management tasks. For example, MixLinear can dynamically forecast future energy usage to optimize operational modes and reduce consumption—applications where existing models like SparseTSF and FITS would be infeasible due to their higher computational and memory demands.
> > > Regarding the specific point about 1D convolutions, we would like to highlight that convolution is a well-established and widely adopted technique in Long-Term Time Series Forecasting (LTSF). As a learnable module, it is commonly used to mitigate high-frequency noise in time series data, allowing for the extraction of meaningful trend information. This approach aligns with established methodologies and is implemented in models such as Temporal Convolutional Networks (TCNs), SparseTSF, and even the paper you kindly recommended, "Forecasting Energy Time Series with Profile Neural Networks" (Section 2.2).
> > > Lastly, we would like to reiterate our commitment to improving the manuscript further. If the reviewer could provide more specific feedback regarding the manuscript's vague or unclear points, we would be delighted to address them directly and thoroughly.
> > > Thank you for your time and input.

---

### Official Review · Reviewer_Xb1P · 2024-10-29

**Soundness:** 2
**Presentation:** 3
**Contribution:** 2
**Rating:** 5
**Confidence:** 4

**Summary:**

This paper introduces a lightweight time series forecasting model that employs trend segmentation, segment transformation, and a parallel approach of trend spectrum compression and trend spectrum transformation. The model aims to maximize performance improvements while minimizing the number of parameters and computational complexity. Through a series of experiments, the authors validated the model's effectiveness.

**Strengths:**

1. The overall approach proposed in this paper is feasible and achieves good results.
2. The authors reduce parameters and computational complexity by employing a two-stage downsampling method (periodic downsampling and trend segmentation).
3. The model demonstrates good transferability and generalization capability.

**Weaknesses:**

1. The foundation of this work is based on trend downsampling, which is the core idea of SparseTSF (ICML 2024 Oral)--namely, periodic downsampling and parameter reuse. The interpolation method in the frequency domain is based on FITS (ICLR 2024 Spotlight). The novelty of this paper is limited.
2. The performance improvements seems to be marginal. Specifically, except for the ETTh1 dataset, the performance on other datasets are not good enough compared to SparseTSF and FITS.
3. The experimental section of this paper lacks sufficient visualization of experimental results. For instance, what's the result of the model's training and convergence compare to other linear models? What is the predictive performance of the model?
4. Although the authors performed ablation studies, they did not explicitly explain the role of key components in the model. For example, what kind of information is learned by the temporal and frequency domain methods? In the temporal transformation, the segment transformation used by the authors is essentially similar to the operation of an MLP-Mixer, which mixes data along two directions. The interpretability of this operation is not clear.
5. The paper lacks a hyperparameter (such as w) search experiment, and the authors did not provide parameter information corresponding to the optimal results across all datasets (they only provided information for Electricity). Is the model's performance sensitive to w across different datasets?
6. In Appendix Table 7, the authors mentioned the filters with n=15 and n=19 points, but the length of the corresponding trend sequence is only 30 (for ETTh1). The FFT of the actual sequence should be conjugate symmetric, meaning it contains only 16 points of valid information. A 19-point filter introduces redundancy and high-frequency noise.
7. Another concern is about the frequency domain transformation. Temporal trend decomposition already removes the inherent periodic and detailed information in the data, which is a method that emphasizes the overall structure. Additionally, performing frequency domain transformations on sequences after periodic downsampling (where w is often large, e.g., 24) will cause severe aliasing effects for most time series. The authors should provide a theoretical analysis or empirical demonstration of how their method addresses or mitigates these aliasing effects.

**Questions:**

1. Regarding W3, more visualized results are expected to be provided.
2. Regarding W4, what's the different features  learned by the frequency and temporal domain methods?
3. Regarding W5, a more detailed introduction of parameter information setting should be added.
4. Regarding W7, the authors should provide a theoretical analysis or empirical demonstration of how their method addresses or mitigates these aliasing effects.
5. Please check Figure 2. In the segement transformation part, the shape should be 4x4 -> 4x5 -> 5x4 -> 5x5. The current shape is 4x4 -> 5x5 -> 5x5 -> 5x5.

---

> ### Author Response · Authors · 2024-11-22
>
> **Comment1:**
>
> The performance improvements seem to be marginal. Specifically, except for the ETTh1 dataset, the performance on other datasets is not good enough compared to SparseTSF and FITS.
>
> **Response:**
> Thank you for your comment. Our key contribution lies in significantly reducing the model size while maintaining competitive prediction performance. Many existing models are too large to be deployed on devices with limited hardware resources. In contrast, our model is designed to be lightweight and suitable for such resource-constrained devices, offering efficient operations without significantly compromising prediction accuracy.
>
> We have incorporated two additional datasets (Exchange and Solar) and two new baselines (iTransformer and SCINet) into our experiments, which are reflected in Table 1 in Section 4.2 (Page 6) of the revised manuscript. The results demonstrate that MixLinear continues to achieve better or comparable prediction performance compared to other baselines.
>
> ---
>
> **Comment2:**
>
> The experimental section of this paper lacks sufficient visualization of experimental results. For instance, what are the training and convergence results of the model compared to other linear models? What is the predictive performance of the model?
>
> **Response:**
> Thank you for your comment. We have added new figures to visualize our experimental results. Specifically:
> - We visualize the training convergence of MixLinear under the input-720-predict-720 setting on the Exchange dataset, comparing it to SparseTSF, FITS, DLinear, PatchTST, and iTransformer. As shown in Figure 3 (Page 20), MixLinear demonstrates faster and smoother convergence compared to other models.
> - We present the predictive performance of MixLinear compared to other baseline models under the input-720-predict-96 and input-720-predict-192 settings on the Exchange dataset. Figures 4 (Page 21) and 5 (Page 22) show that MixLinear consistently outperforms all baselines.
>
> This analysis has been added to Appendix B.7 and B.8 of the revised manuscript.
>
> ---
>
> **Comment3:**
>
> The paper lacks a hyperparameter (such as $ w $) search experiment, and the authors did not provide parameter information corresponding to the optimal results across all datasets (only provided for Electricity). Is the model's performance sensitive to $ w $ across different datasets?
>
> **Response:**
> Thank you for your feedback. To evaluate the effect of the period $ w $ used for downsampling, we conducted new experiments by varying $ w $ from 2 to 36 across forecasting horizons of 96, 192, 336, and 720. We added the results to Appendix B.5, Table 10 (Page 18) of the revised manuscript.
>
> For most datasets, $ w = 24 $ achieves the best performance. However, on the Exchange dataset (financial data that lacks strong seasonality), smaller values of $ w $ (e.g., $ w = 2 $) yield better results, suggesting a different temporal structure where shorter periods are more suitable.
>
> ---
>
> **Comment4:**
>
> In Appendix Table 7, the authors mentioned filters with $ n = 15 $ and $ n = 19 $ points, but the length of the corresponding trend sequence is only 30 (for ETTh1). The FFT of the actual sequence should be conjugate symmetric, meaning it contains only 16 points of valid information. A 19-point filter introduces redundancy and high-frequency noise.
>
> **Response:**
> Thank you for your feedback. We performed additional experiments with MixLinear using different low-pass filter (LPF) values. While an LPF greater than 15 may introduce redundancy and high-frequency noise due to components from negative frequencies, our experiments show that in certain cases, an LPF greater than 15 can still yield better performance.
>
> The new results have been added to Appendix B.2, Table 7 (Page 16) of the revised manuscript.
>
> ---

---

> ### Author Response · Authors · 2024-11-22
>
> **Comment5:**
>
> Another concern is about the frequency domain transformation. Temporal trend decomposition already removes the inherent periodic and detailed information in the data, emphasizing the overall structure. Additionally, performing frequency domain transformations on sequences after periodic downsampling (where $ w $ is often large, e.g., 24) may cause severe aliasing effects for most time series. The authors should provide a theoretical analysis or empirical demonstration of how their method addresses or mitigates these aliasing effects.
>
> **Response:**
> Thank you for your comment. While periodic downsampling with a large $ w $ (e.g., 24) can lead to severe aliasing effects, our approach mitigates this issue through an aggregation step performed before downsampling. Specifically:
> - We apply a 1D convolution with kernel size $ w $, aggregating all information within each period at every time step.
> - After aggregation, we downsample the series by the period $ w $, resulting in the trend component.
>
> This approach effectively decouples the periodic and trend components, producing a compact representation where each trend time point encapsulates the complete information from one period. This method significantly reduces the risk of aliasing while preserving essential characteristics of the time series.
>
> ---
>
> **Comment6:**
>
> Although the authors performed ablation studies, they did not explicitly explain the role of key components in the model. For example, what kind of information is learned by the temporal and frequency domain methods? In the temporal transformation, the segment transformation used by the authors is essentially similar to the operation of an MLP-Mixer, which mixes data along two directions. The interpretability of this operation is not clear.
>
> **Response:**
> Thank you for your feedback. In the time domain, MixLinear captures intra-segment and inter-segment variations by decoupling channel and periodic information from trend components, breaking the trend into smaller, manageable segments. In the frequency domain, the model captures variations by mapping these decoupled subsequences into a latent frequency space and reconstructing the trend spectrum effectively.
>
> While MixLinear shares some similarities with MLP-Mixer, it differs significantly in its purpose and design. The two linear layers of MixLinear in the time domain transformation serve to minimize the model scale:
> - The **intra-segment linear layer** captures short-term patterns within segments.
> - The **inter-segment linear layer** captures long-term patterns across segments.
>
> Together, these layers enable MixLinear to balance local (short-term) and global (long-term) feature extraction, providing a comprehensive representation of the temporal structure.

---

> ### Comment · Reviewer_Xb1P · 2024-11-24
> **Thank you for your response**
>
> Thank you to the authors for their efforts in addressing the issues I raised. The experimental concerns I mentioned have been resolved. Overall, as I previously noted, the proposed model is largely based on existing lightweight models, merely further segmenting the trend components and performing intra-segment and inter-segment mixing. However, the authors did not attempt to use theoretical explanations to demonstrate the effectiveness of the method, and there is a lack of intuitive visualization of the roles of the various components (trend extraction, time-domain, and frequency-domain processing). This makes the integration of these components somewhat like a mechanical patchwork. Furthermore, although the MixLinear method effectively reduces the parameter count, its unstable results (the model's performance is not always guaranteed to be the best or even the second best) make its advantages over SparseTSF, FITS, and similar methods rather limited. I am inclined to maintain the original score.

---

> > ### Author Response · Authors · 2024-11-27
> > **Misunderstanding on performance of MixLinear**
> >
> > > its unstable results (the model's performance is not always guaranteed to be the best or even the second best) make its advantages over SparseTSF, FITS, and similar methods rather limited.
> >
> > MixLinear (0.1K) achieves state-of-the-art performance in low-channel prediction scenarios, such as ETTh1, ETTh2, and Exchange, solidifying its position as the best model for these tasks.
> >
> > In high-channel scenarios like Electricity, Traffic, and Solar, MixLinear also performs competitively. For the longest prediction horizon (720 steps), MixLinear consistently ranks as the second-best model, with an MSE difference of less than 0.005 compared to SparseTSF and FITS. This demonstrates that MixLinear maintains its efficiency and accuracy even in challenging high-channel forecasting tasks.

---

> > ### Author Response · Authors · 2024-11-27
> > **Visualization**
> >
> > Intuitive visualization of the roles of the various components
> >
> > We have included extensive visualizations in **Appendix C (Pages 24–27)**, which provide a detailed analysis and comparison of the weights and prediction patterns of our model. These visualizations highlight the following insights:
> >
> > 1. **Comparison with Other Models**:
> >    Figures **6** and **8** vividly demonstrate that our model effectively compresses the most focused components of DLinear (40K) and SparseTSF (1K) into a **smaller space in the time domain**. Similarly, it compresses the most focused components of FITS (10K) into a **lower-dimensional space in the frequency domain**, showcasing the efficiency and adaptability of MixLinear.
> >
> > 2. **Time Domain Weights**:
> >    Figure **7** illustrates the **intra-segment** and **inter-segment weights** of MixLinear in the time domain. The intra-segment weights primarily focus on capturing **local patterns**, while the inter-segment weights emphasize **global patterns**, effectively balancing detailed local features with broader temporal trends.
> >
> > 3. **Frequency Domain Weights**:
> >    - Figure **9a** shows that MixLinear compresses frequency components into a **frequency domain latent space**, enabling it to learn different **filtering patterns**:
> >      - The **first dimension** of the real part focuses on **high-frequency components**, capturing intricate details from both **positive and negative frequencies**.
> >      - The **second dimension** of the real part emphasizes **low-frequency components**, representing broader trends.
> >    - Figure **9b** highlights that these two dimensions capture **distinct filtering patterns**, further illustrating the flexibility of MixLinear in processing frequency-domain features.
> >    - Figures **9c** and **9d** show that the **imaginary part (phase information)** of the weights captures **clearer and more complex patterns**, demonstrating its role in constructing a more nuanced spectral representation.
> >
> > 4. **Prediction Patterns in Time and Frequency Domains**:
> >    Figures **10a** and **10b** compare the predictions of MixLinear's **time domain** and **frequency domain modules** on two data fragments from ETTh1. The **frequency domain predictions** appear smoother due to the application of filters, while the **time domain predictions** capture finer details, emphasizing localized variations. This complementary behavior highlights the effectiveness of MixLinear in leveraging both domains for improved forecasting.
> >
> > These visualizations provide a more intuitive understanding of the roles of different components in MixLinear, showcasing its ability to efficiently capture and process patterns in both time and frequency domains.

---

> > ### Author Response · Authors · 2024-11-27
> > **Difference between Spectrum Compression (MixLinear) and interpolation(FITS)**
> >
> > >The interpolation method in the frequency domain is based on FITS (ICLR 2024 Spotlight)
> >
> > The interpolation used in FITS directly applies a **low-pass filter** on the spectrum and performs interpolation on the raw frequency components. However, this approach overlooks high-frequency details, and the rFFT it employs completely ignores the contributions from **negative frequency components**.
> >
> > In contrast, the **spectrum compression** used in our model, MixLinear, compresses the frequency components into a **lower-dimensional latent space**, enabling it to learn diverse filtering patterns. This adaptability makes MixLinear more effective for **long time series data forecasting**.

---

> ### Author Response · Authors · 2024-11-27
> **Misunderstanding on the Novelty of MixLinear**
>
> >merely further segmenting the trend components and performing intra-segment and inter-segment mixing. This makes the integration of these components somewhat like a mechanical patchwork.
>
> The key idea behind SparseTSF is its Cross-Period Sparse Forecasting technique, which focuses on downsampling original sequences to capture cross-period trend predictions. It does not involve segmenting.
>
> In contrast, the core concept of MixLinear in the time domain lies in segmenting the trends and performing both intra-segment and inter-segment transformations. This approach enables MixLinear to effectively capture both local and global patterns within the data. While MixLinear employs downsampling as a preprocessing step—leveraging SparseTSF techniques—it builds on this foundation with additional segmentation-based transformations.
>
> In the frequency domain, FITS primarily focuses on positive frequency components and employs a low-pass filter (LPF) to eliminate high-frequency noise. However, our approach incorporates both positive and negative frequency components, which has proven effective in enhancing prediction performance.
>
> Furthermore, we compress the frequency components into a 2D frequency latent space. As illustrated in Figure 9a (Page 26), the first dimension of the weights predominantly captures high-frequency components, while the second dimension focuses on low-frequency components. This demonstrates that our method is more adaptive than FITS in capturing complex patterns within the frequency domain.

---

> ### Author Response · Authors · 2024-11-27
> **Difference between Downsampling and Segmentation in Computation Complexity**
>
> >The foundation of this work is rooted in **trend downsampling**, which is the core idea behind SparseTSF (ICML 2024 Oral)—specifically, **periodic downsampling** and **parameter reuse**.
>
> MixLinear differs significantly, not only in the **patterns it learns** but also in its **computational complexity**.
>
> For a time series of length $L$ with a period $w$:
> - The computational complexity for **downsampling** (as used in SparseTSF) is $O\left(\frac{L}{w} \cdot \frac{L}{w}\right) = O\left(\left(\frac{L}{w}\right)^2\right) = O(L^2)$.
> - In contrast, the computational complexity for **segmentation** (as used in MixLinear) is $O\left(\sqrt{L} \cdot \sqrt{L} + \sqrt{L} \cdot \sqrt{L}\right) = O(L)$.
>
> This significant reduction in complexity gives MixLinear a clear advantage over SparseTS

---

### Official Review · Reviewer_Atcp · 2024-11-01

**Soundness:** 3
**Presentation:** 3
**Contribution:** 2
**Rating:** 5
**Confidence:** 4

**Summary:**

This paper propose MixLinear, an ultra-lightweight multivariate time series forecasting model which is specifically designed for source-constrained devices. The proposed MixLinear can effectively captures both temporal and frequency domain features with lower complexity. Meanwhile, MixLinear only needs 0.1k parameters with comparable performance.

**Strengths:**

1. MixLinear has comparable performance with only 0.1k parameters.
2. The proposed MixLinear is the first lightweight model that captures both temporal and frequency domain information.
3. MixLinear has little running time compared to other time series forecasting models.
4. The model is simple and easy to follow.

**Weaknesses:**

1.	The authors demonstrate the computational complexity of MixLinear is $O(n)$, but additional operations such as Fourier Transform introduce additional complexity. A more detailed discussion on the computational complexity of MixLinear would be better.
2.	In table 2, the authors report the running time of these models during, However, it would be better to report the running time in the inference phase.
3.	The experimental results in Tables 1 and 6 show that MixLinear performs competitively only on the ETTh1 and ETTh2 datasets, while its performance lags behind state-of-the-art models on other datasets.
4.	Regarding to the number of parameters, although MixLinear needs only 0.1k parameters, I am curious about what kind of lightweight device cannot store around 10k parameters. If such a device exists, is it meaningful to use it for time series forecasting?
5.	Also, even though MixLinear has a small parameter count, I believe its computational complexity is not low. And I think the authors need to discuss the computational complexity in more depth.
6.	The superscripts and subscripts in Figure 2 need to be adjusted.

**Questions:**

In Figure 2, I noticed that MixLinear seems to only utilize the trend component while discarding the periodic component. I would like to ask the authors why the periodic component was discarded. Are there relevant experimental results that demonstrate that using only the trend component is more effective?

**Details Of Ethics Concerns:**

There is no ethics concerns.

---

> ### Author Response · Authors · 2024-11-22
>
> **Comment1:**
> The authors demonstrate the computational complexity of MixLinear is $ O(n) $, but additional operations such as Fourier Transform introduce additional complexity. A more detailed discussion on the computational complexity of MixLinear would be better.
>
> **Response:**
> Thank you for your comment. We have demonstrated in the paper that the computational complexity of MixLinear is $ O(n) $. For model computation complexity analysis, MixLinear models long-term dependencies in the frequency domain by mapping the decoupled time series subsequences (trends) into a latent frequency space, where it reconstructs the trend spectrum. To further enhance efficiency, MixLinear applies an aggregation downsampling step using a 1D CNN with a kernel size of $ w $, resulting in a computational complexity of $ O(w) $. Following this, in the frequency domain transformation, FFT and iFFT operations are applied to the downsampled trend segments of length $ n $, introducing an additional time complexity of $ O(n \log n) $ and space complexity of $ O(n) $.
>
> In summary:
> - Total time complexity of MixLinear: $ O(w) + O(n \log n) + O(n) $, which simplifies to $ O(n \log n) $.
> - Total space complexity of MixLinear: $ O(w) + O(n) + O(n) $, which simplifies to $ O(n) $.
>
> We have added this analysis to Appendix B.6 (Page 19) of the revised manuscript.
>
> ---
>
>
> **Comment2:**
> In Table 2, the authors report the running time of these models during training. However, it would be better to report the running time in the inference phase.
>
> **Response:**
> Thank you for your comment. We have measured the inference time for MixLinear, which is 0.6 ms, the smallest among all methods. We have added these measurements to Table 2 in Section 4.3 (Page 7) of the revised manuscript.
>
>
>
> **Commen3:**
> The experimental results in Tables 1 and 6 show that MixLinear performs competitively only on the ETTh1 and ETTh2 datasets, while its performance lags behind state-of-the-art models on other datasets.
>
> **Response:**
> Thank you for your comment. Our key contribution lies in significantly reducing the model size while maintaining competitive prediction performance. Many existing models are too large for deployment on end devices with limited hardware resources. In contrast, MixLinear is specifically designed to be lightweight and suitable for resource-limited devices. It provides efficient operations without compromising much on prediction accuracy.
>
> To address your concern, we have added two more datasets (Exchange and Solar) and two additional baselines (ITransformer and SCINet) to our experiments. MixLinear continues to deliver top 2 performance on 4 out of the 6 datasets. We have updated Table 1 in Section 4.2 (Page 6) of the revised manuscript.
>
> ---
>
>
> **Comment4:**
> Even though MixLinear has a small parameter count, its computational complexity does not seem low. The authors need to discuss the computational complexity in more depth.
>
> **Response:**
> Thank you for your feedback. MixLinear models long-term dependencies by mapping decoupled time series subsequences (trends) into a latent frequency space, reconstructing the trend spectrum for effective representation of periodic variations. To improve efficiency:
> - It uses a 1D CNN with kernel size $ w $ for aggregation and downsampling, with complexity $ O(w) $.
> - FFT and iFFT operations are applied to the downsampled segments of length $ n $, with time complexity $ O(n \log n) $ and space complexity $ O(n) $.
>
> In summary:
> - Total time complexity: $ O(w) + O(n \log n) + O(n) $, simplified to $ O(n \log n) $.
> - Total space complexity: $ O(w) + 2O(n) $, simplified to $ O(n) $.
>
> We have included a detailed analysis in Appendix B.6 (Page 19) of the revised manuscript.
>
> ---
>
>
> **Comment5:**
> The superscripts and subscripts in Figure 2 need to be adjusted.
>
> **Response:**
> Thank you for your comment. We have revised Figure 2 in Section 3.1 (Page 4) of the revised manuscript.
>
> ---
>
> **Comment6:**
> In Figure 2, it seems that MixLinear only utilizes the trend component while discarding the periodic component. Why was the periodic component discarded? Are there experimental results showing that using only the trend component is more effective?
>
> **Response:**
> Thank you for your feedback. The periodic components are not discarded. We perform aggregation before downsampling. For aggregation, we apply a 1D convolution with a kernel size $ w $, which aggregates all information within each period at every time step. We then downsample the aggregated series by the period $ w $, resulting in the trend component $ X_{\text{Trend}} \in \mathbb{R}^{n} $, where $ n = \left\lceil \frac{L}{w} \right\rceil $. This method effectively decouples the periodic and trend components, providing a compact representation where each trend time point encapsulates all the information from one period in the original series.

---

> ### Comment · Reviewer_Atcp · 2024-11-27
>
> Thanks for the author's responses to my questions. I will keep my original score and will consider revising it after discussing with other reviewers.

---

> > ### Author Response · Authors · 2024-11-27
> >
> > Dear Reviewer,
> >
> > Thank you for taking the time to review our work and provide your valuable feedback. We welcome any additional comments, questions, or concerns you may have, as it would give us an opportunity to address them further.

---

### Official Review · Reviewer_rx7j · 2024-11-03

**Soundness:** 2
**Presentation:** 2
**Contribution:** 2
**Rating:** 5
**Confidence:** 3

**Summary:**

This paper introduces MixLinear, an ultra-lightweight model for long-term time series forecasting (LTSF) that achieves comparable or better accuracy than state-of-the-art models while using only 0.1K parameters. The key innovation of MixLinear lies in its unique approach of capturing features from both time and frequency domains. In the time domain, it applies trend segmentation to capture intra-segment and inter-segment variations, while in the frequency domain, it reconstructs trend spectrums from a low-dimensional latent space. By reducing the parameter scale from O(n²) to O(n) for n-length inputs/outputs, MixLinear achieves efficient computation without sacrificing accuracy. The authors demonstrate MixLinear's effectiveness through extensive experiments on four benchmark datasets, showing it can achieve up to 5.3% reduction in Mean Squared Error compared to state-of-the-art models. The paper also demonstrates MixLinear's strong generalization capability across different datasets and its robustness in various forecasting scenarios, making it particularly suitable for deployment on resource-constrained devices.

**Strengths:**

The paper shows good originality by being the first to effectively combine time and frequency domain features in a lightweight LTSF model - a creative integration that differs from conventional single-domain approaches. The technical quality is demonstrated through experiments across multiple benchmarks, convincingly showing that MixLinear achieves comparable or better accuracy with just 0.1K parameters versus 1K-6M in baselines. The paper is clearly written. The work makes a good contribution by enabling accurate LTSF deployment on resource-constrained devices, while providing valuable insights about combining domain features efficiently.

**Weaknesses:**

- There are much more time series baselines to be compared with.
- The number and kinds of datasets that the experiments are based on are too limited, totally 4 and 3 of them are electricity-related datasets.

**Questions:**

For table 4, can you also show ETTh1 -> ETTh2, ETTh1 -> Electricity?

---

> ### Author Response · Authors · 2024-11-22
>
> ### Comment 1
>
> *There are many more time series baselines to be compared with. The number and kinds of datasets that the experiments are based on are too limited, totally 4 and 3 of them are electricity-related datasets.*
>
> **Response:**
> Thank you for your comment. We have applied our MixLinear and baselines on two more datasets (*Exchange* and *Solar*) and examined their performance. Below are the results. We have also added two baselines (*ITransformer* and *SCINet*). MixLinear consistently provides small MSE and outperforms these new baselines. The updated experimental results have been added to **Table 1** in **Section 4.2 on Page 6** of the revised manuscript.
>
> | **Models**          | **MixLinear (Ours)** | **SparseTSF (2024)** | **FITS (2024)** | **DLinear (2023)** | **PatchTST (2023)** | **ITransformer (2023)** | **SCINet (2022)** | **TimesNet (2022)** | **FEDformer (2022)** | **Diff.** |
> |----------------------|----------------------|-----------------------|-----------------|--------------------|---------------------|-------------------------|-------------------|---------------------|----------------------|-----------|
> | **Solar** - 96      | **0.211**           | **0.211**             | **0.195**       | 0.290             | 0.265               | **0.203**              | 0.237             | 0.373               | 0.286               | -0.017    |
> | **Solar** - 192     | **0.227**           | **0.225**             | **0.216**       | 0.320             | 0.288               | 0.233                  | 0.280             | 0.397               | 0.291               | -0.011    |
> | **Solar** - 336     | **0.240**           | **0.241**             | **0.232**       | 0.353             | 0.301               | 0.248                  | 0.304             | 0.420               | 0.354               | -0.008    |
> | **Solar** - 720     | **0.240**           | **0.241**             | **0.242**       | 0.357             | 0.295               | 0.249                  | 0.308             | 0.420               | 0.380               | +0.001    |
> | **Exchange** - 96   | **0.088**           | 0.105                 | **0.086**       | **0.087**         | **0.087**           | **0.086**              | 0.267             | 0.107               | 0.148               | -0.002    |
> | **Exchange** - 192  | **0.175**           | 0.196                 | **0.180**       | 0.251             | 0.183               | **0.177**              | 0.351             | 0.226               | 0.271               | +0.002    |
> | **Exchange** - 336  | **0.318**           | 0.358                 | **0.333**       | 0.403             | 0.390               | **0.331**              | 0.424             | 0.367               | 0.460               | +0.013    |
> | **Exchange** - 720  | **0.923**           | **0.954**             | **0.941**       | 1.364             | 1.038               | 0.970                  | 1.058             | 0.964               | 1.195               | +0.018    |
>
> ---

---

> > ### Author Response · Authors · 2024-11-22
> >
> > ### Comment 2
> >
> > *For Table 4, can you also show ETTh1 → ETTh2, ETTh1 → Electricity?*
> >
> > **Response:**
> > Thank you for your comment. We have added four new sets of experiments to evaluate the generalization of different methods, including *ETTh1 → ETTh2* and *ETTh1 → Electricity*. Below is the updated performance summary. While SparseTSF performs competitively in some cases, MixLinear generally outperforms it, especially in tasks involving significant domain shifts, such as *Exchange → ETTh2* and *Solar → ETTh2*. FiTS and DLinear exhibit moderate performance but consistently fall behind MixLinear, particularly for longer horizons like 336 and 720. PatchTST, though strong in certain scenarios, does not match MixLinear's overall robustness and generalization ability. The new experimental results have been added to **Table 9 in Appendix B.4 on Page 17** of the revised manuscript.
> >
> > | **Dataset**                     | **Horizon** | **Metrics** | **MixLinear** | **SparseTSF** | **FITS** | **DLinear** | **PatchTST** |
> > |----------------------------------|-------------|-------------|---------------|---------------|----------|-------------|--------------|
> > | **Exchange → ETTh2**            | 96          | MSE         | **0.406**     | 0.413         | 0.444    | 0.478       | 0.459        |
> > |                                  | 192         | MSE         | **0.507**     | 0.515         | 0.532    | 0.475       | 0.573        |
> > |                                  | 336         | MSE         | **0.542**     | 0.607         | 0.581    | 0.768       | 0.617        |
> > |                                  | 720         | MSE         | 0.600         | 0.582         | 0.600    | 1.825       | **0.556**    |
> > | **Solar → ETTh2**               | 96          | MSE         | **0.369**     | 0.369         | 0.371    | 0.433       | 0.503        |
> > |                                  | 192         | MSE         | 0.388         | 0.384         | 0.374    | 0.901       | 0.537        |
> > |                                  | 336         | MSE         | **0.397**     | 0.398         | 0.398    | 0.818       | 0.446        |
> > |                                  | 720         | MSE         | 0.422         | 0.423         | **0.419**| 0.976       | 0.536        |
> > | **ETTh1 → ETTh2**               | 96          | MSE         | 0.290         | **0.293**     | **0.282**| 0.296       | 0.389        |
> > |                                  | 192         | MSE         | 0.352         | **0.353**     | 0.349    | 0.378       | 0.365        |
> > |                                  | 336         | MSE         | 0.374         | **0.373**     | 0.376    | 0.436       | 0.494        |
> > |                                  | 720         | MSE         | **0.388**     | 0.390         | 0.390    | 0.627       | 0.444        |
> > | **ETTh1 → Electricity**         | 96          | MSE         | 0.326         | 0.326         | **0.309**| 0.325       | 0.560        |
> > |                                  | 192         | MSE         | 0.377         | 0.374         | **0.363**| 0.371       | 0.554        |
> > |                                  | 336         | MSE         | 0.390         | 0.392         | **0.383**| 0.427       | 0.711        |
> > |                                  | 720         | MSE         | 0.404         | 0.404         | **0.401**| 0.570       | 0.812        |

---

> ### Comment · Reviewer_rx7j · 2024-11-28
>
> Thanks for the additional experiment results attached by the author. Your method did not consistently outperform the baseline methods, especially the FITS method. Thus, I think the performance is not sufficiently impressive and I will maintain my score.

---

### Meta-Review · Area_Chair_Qquo · 2024-12-19

**Metareview:**

This paper introduces a ultra-lightweight linear model for long-term forecasting that captures  temporal and frequency domain features using only 0.1K parameters. The model, while  intuitive, does not distinguish itself purely on novelty (as some reviewers pointed out), given that it is largely based on lightweight linear models that prior works have also used (albeit in a different way). This requires the paper to perform a much stronger empirical evaluation than what it currently has. While the paper reports strong results on the baselines considered, evaluating it on 6 datasets (as another reviewer pointed out) is simply not sufficient, given that the time-series community now has much more comprehensive datasets and benchmarks. I would urge the authors to conduct a more in-depth evaluation of the model's performance on broader and more diverse benchmarks (given its low-resource constraints, it does not need to outperform all SOTA baselines on accuracy, but simply show competitive results), and resubmit this work to a future venue.

**Additional Comments On Reviewer Discussion:**

Reviewers asked for the authors to add more datasets and baselines, and also add inference time complexity measurements and more experimental visualizations. While the authors did add some more datasets and baselines,  the improved experimental section still is not comprehensive enough to meet the bar for ICLR (especially for a paper whose merits rely more on strong empirical performance and less on novelty)

---

### Decision · Program_Chairs · 2025-01-22

Reject